# Budding yeast Rif1 binds to replication origins and protects DNA at blocked replication forks

Shin-ichiro Hiraga[1],[*] iD, Chandre Monerawela[1], Yuki Katou[2], Sophie Shaw[3] iD, Kate RM Clark[1], Katsuhiko Shirahige[2] & Anne D Donaldson[1],[**] iD

## Abstract

Despite its evolutionarily conserved function in controlling DNA replication, the chromosomal binding sites of the budding yeast Rif1 protein are not well understood. Here, we analyse genome-wide binding of budding yeast Rif1 by chromatin immunoprecipitation, during G1 phase and in S phase with replication progressing normally or blocked by hydroxyurea. Rif1 associates strongly with telomeres through interaction with Rap1. By comparing genomic binding of wild-type Rif1 and truncated Rif1 lacking the Rap1-interaction domain, we identify hundreds of Rap1-dependent and Rap1-independent chromosome interaction sites. Rif1 binds to centromeres, highly transcribed genes and replication origins in a Rap1-independent manner, associating with both early and late-initiating origins. Interestingly, Rif1 also binds around activated origins when replication progression is blocked by hydroxyurea, suggesting association with blocked forks. Using nascent DNA labelling and DNA combing techniques, we find that in cells treated with hydroxyurea, yeast Rif1 stabilises recently synthesised DNA. Our results indicate that, in addition to controlling DNA replication initiation, budding yeast Rif1 plays an ongoing role after initiation and controls events at blocked replication forks.

**Keywords** centromere; ChIP-Seq; DNA replication origin; nascent DNA; Rif1
**Subject Category** DNA Replication, Repair & Recombination

## Introduction

Chromosomes are highly dynamic, and chromatin changes its structural composition during functional processes and at different cell cycle stages. For example, replication or transcription fork passage requires the disassembly then reassembly of nucleosomes. During mitosis, chromatin is compacted and must withstand the physical tension occurring during sister chromatid segregation. Additionally, spontaneous and replication-associated damage to chromosomes must be repaired in a timely way during the cell cycle. Failure or incomplete execution of any of these processes can cause genome instability.

Rif1 is an evolutionarily conserved protein involved in multiple genome integrity pathways. Rif1 was originally identified in the budding yeast *Saccharomyces cerevisiae* as a component of telomeric chromatin that regulates telomere length [1,2]. Rif1 associates with telomeres, and with the *MAT* locus and mating cassettes, mainly through interaction with the transcription factor Rap1 [3]. Rap1 recognises a TG-rich motif present upstream of genes it regulates. This recognition motif also occurs within the telomeric terminal TG repeat sequences, and multiple copies of Rap1 bind telomeres [4]. While originally isolated for its role in binding Rap1 at telomeres, Rif1 (Rap1-Interacting Factor 1) has recently been identified as an important regulator of DNA replication initiation, in a function conserved from yeast to human [5–11]. Despite replication control being one of its conserved functions, it however proved difficult for many years to demonstrate binding of Rif1 at replication origin sites. Rif1 has been implicated in additional pathways of genome integrity, in particular directing double-strand break repair pathway choice [12–18], and suppressing or resolving mitotic chromosome entanglements [19,20].

A critical step in replication initiation is executed by Dbf4-dependent protein kinase (DDK), which promotes DNA replication initiation by phosphorylating the MCM complex to activate it as the replicative helicase. In DNA replication control, Rif1 counteracts the function of DDK by directing Protein Phosphatase 1 (PP1) to dephosphorylate MCM proteins and oppose replication initiation. Notably, this action of Rif1 as a substrate-targeting subunit for PP1 is evolutionarily conserved, with Rif1 also controlling replication initiation in mammalian cells [8–10,21–23]. Budding yeast Rif1 and PP1 bound to the chromosome ends specify the late replication timing of origins in the vicinity of telomeres [8,10,24–26]. However, in *S. cerevisiae*, the importance of Rif1 for the replication timing programme at sites other than telomeres appears to be fairly minor in an unimpeded S phase [26]. Although its contribution to specification of the replication temporal programme occurs primarily at telomeres, budding yeast Rif1 does nonetheless clearly affect initiation at numerous replication origins genome-wide, since when available DDK is

1 Institute of Medical Sciences, University of Aberdeen, Aberdeen, UK
2 Institute for Quantitative Biosciences, University of Tokyo, Tokyo, Japan
3 Centre for Genome-Enabled Biology and Medicine, University of Aberdeen, Aberdeen, UK
*Corresponding author. Tel: +44 1224 437317; E-mail: s.hiraga@abdn.ac.uk
**Corresponding author. Tel: +44 1224 437316; E-mail: a.d.donaldson@abdn.ac.uk

limited, deletion of *RIF1* permits replication initiation to occur throughout large tracts of the genome [9]. Also, when replication is blocked by hydroxyurea (HU), additional replication origins show initiation in a *rif1Δ* mutant when compared to wild type [11]. These observations suggest that Rif1 does play a critical role in controlling origin firing, especially under replication stress, even though the role of *S. cerevisiae* Rif1 in normal replication timing control otherwise appears largely limited to telomeric regions.

Our understanding of the effects of Rif1 in replication control has been impeded by the fact that it has been difficult to detect Rif1 interacting directly with replication origins, even in *S. cerevisiae* which has the best understood replication origin sites of any eukaryote [27,28]. No high-resolution chromatin immunoprecipitation (ChIP) analysis has been described for *S. cerevisiae* Rif1, and until now, the available information describing chromosome association of Rif1 in budding yeast has been limited to a few very specific sites, including the telomeres, the *MAT* locus, and mating type cassettes—interactions all mediated mainly through interaction with Rap1 [3]. One impediment to genome-wide analysis of Rif1 binding has been the very strong preference displayed by Rif1 for these specific, repeated chromosomal loci, which tended to obscure binding to other loci in microarray analyses of ChIP experiments. To obtain a better understanding of Rif1 function in DNA replication and genome maintenance, we examined the chromatin association patterns of Rif1 by next-generation sequencing analysis of ChIP samples (ChIP-Seq), which provides an improved dynamic range of analysis compared to microarrays. As well as wild-type Rif1, we examined a mutated version of Rif1 that lacks the Rap1-interaction domain, to distinguish Rap1-dependent and Rap1-independent binding sites. Our results reveal Rif1 interaction with several new classes of chromosome loci. We find clear association of Rif1 with replication origins throughout the yeast genome. We additionally detect Rif1 localised to various other types of genomic site, including blocked replication forks, highly transcribed genes and centromeric sequences. These observations suggest potential new roles for Rif1 in modulating chromosome transactions. Investigating in particular the role of Rif1 at blocked replication forks, we find that at forks whose progression is blocked by hydroxyurea treatment, Rif1 is crucial to protect newly replicated DNA.

## Results

### The C-terminal portion of yeast Rif1 is dispensable for opposing DDK in replication control

Rif1 associates with telomeric regions, consistent with its function in controlling telomere length and replication timing near telomeres [1,2,8,11,25]. Rif1 additionally binds the *HML*, *HMR* and *MAT* loci, interacting with Rap1 at these sites as at telomeres [3,29]. However, although it impacts on replication control more broadly, the association of budding yeast Rif1 protein with other chromosomal loci has not been reported. We suspected that its strong preference for telomeric regions might have hindered the detection of Rif1 at non-telomeric regions in previous microarray studies. Structural studies revealed two domains within Rif1 that interact with Rap1: the Rap1-binding motif (RBM) and a C-terminal domain (CTD; Fig 1A) [30]. To explore the behaviour and physiological functions of Rif1

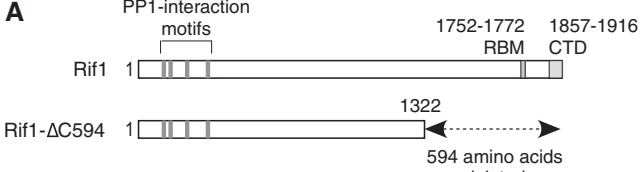

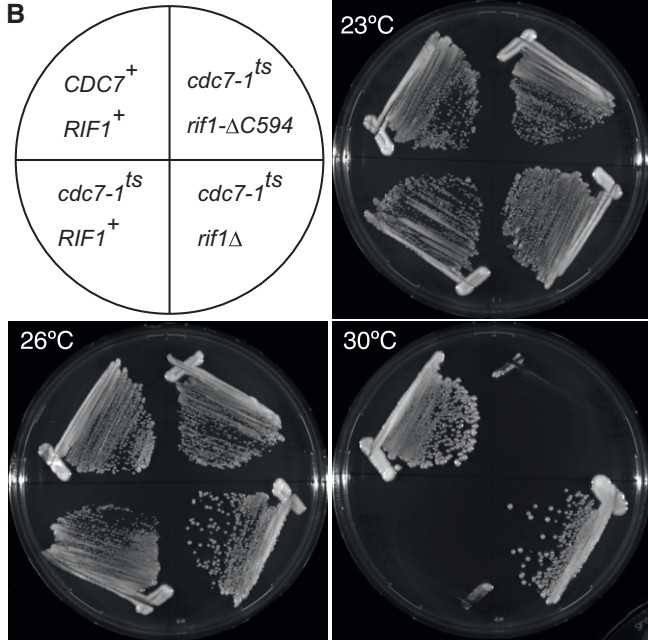

**Figure 1. The C-terminus of yeast Rif1 is dispensable for control of DNA replication.**

A   Structure of budding yeast Rif1 and C-terminally truncated mutant Rif1-ΔC594. RBM, Rap1-binding motif; CTD, C-terminal domain.

B   Rif1-ΔC594 retains function to control DNA replication. Growth of a *cdc7-1 rif1-ΔC594* mutant was compared with growth of *cdc7-1 RIF1* and *cdc7-1 rif1Δ* strains at 23°C (permissive temperature for *cdc7-1* allele), 26°C (mild restrictive temperature) and 30°C (strict restrictive temperature).

independent of Rap1, we used a truncated version of Rif1 lacking the RBM and CTD (Fig 1A) [10]. This C-terminally truncated *RIF1* allele, *rif1-ΔC594*, retains the ability to control DNA replication by counteracting DDK, since it represses growth of a *cdc7-1* mutant strain at 30°C, like wild-type *RIF1⁺*. A *rif1Δ* allele in contrast permits growth of a *cdc7-1* mutant at 30°C (Fig 1B), as previously described. The repressive effect of *rif1-ΔC594* on *cdc7-1* growth is consistent with previous observations that the C-terminal region of Rif1 is dispensable for replication control [9,10]. We designate the truncated protein Rif1-ΔC594.

### ChIP-Seq analysis identifies Rif1 genomic binding site dependent on Rap1

Since next-generation sequencing provides an increased dynamic range compared to microarray-based methods, we investigated whether ChIP-Seq analysis could reveal non-telomeric chromosome association sites of Rif1. We also tested binding of Rif1-ΔC594, to distinguish Rap1-dependent and Rap1-independent chromosome

association sites. Myc-tagged versions of Rif1 and Rif1-ΔC594 were utilised to enable ChIP-Seq analysis of chromatin association. Binding was analysed during G1 phase (cells blocked with α-factor), 60 and 90 min after release from α-factor at 16°C, and in cells released from α-factor into hydroxyurea (HU) to block replication fork progression. As expected [3], full-length Rif1 showed strong binding to telomeres (Fig 2A blue plots), as well as mating type cassettes (Fig EV1A blue plots). Rif1-ΔC594, in contrast, showed greatly reduced association with telomere and subtelomeric sequences (Fig 2A red plots) and virtually no association with mating type loci (Fig EV1A red plots). These effects on association with telomeres and mating loci are as expected, since both modes of binding depend to a large extent on interaction with Rap1.

As well as associating with telomeres, Rap1 regulates multiple genes as a promoter-bound transcription factor. At these sites of Rap1 transcriptional control, we detected binding of Rif1 dependent on its C-terminal Rap1 interaction domain. For example, full-length Rif1 bound the promoter regions of the Rap1-regulated genes *PAU3* and *MAM3* (Fig 2B, blue plots) [31–33]. In contrast, Rif1-ΔC594 protein did not bind these promoters (Fig 2B, red plots). Our results therefore clearly demonstrate that Rif1-ΔC594 is defective for Rap1-dependent association, both with telomeres and sites of Rap1-mediated transcription regulation.

### ChIP-Seq analysis identifies multiple Rap1-independent Rif1 genomic binding sites

In addition to the expected Rap1-dependent associations, we observed that full-length Rif1 and Rif1-ΔC594 bind hundreds of additional sites, with binding intensity often appearing higher for the Rif1-ΔC594 protein.

Both full-length Rif1 and Rif1-ΔC594 associate with many DNA replication origins (Figs 2A, and 3A and B), including both telomere-proximal origins and origins distant from telomeres, as discussed in more detail below.

Unexpectedly, we also found Rif1 and Rif1-ΔC594 association with the coding regions of highly transcribed genes, for example *ACT1*, *RPL22B* and *HAC1* genes as shown in Fig 2A, and tRNA genes as illustrated by Fig 3A. Since Rif1-ΔC594 binds to these sites with generally similar intensity to full-length Rif1, association with highly transcribed loci appears to be independent of Rap1 interaction.

We also noticed association of Rif1 with centromeres (Figs 3C and EV2A). Full-length Rif1 binds some centromeres fairly weakly during G1 phase, but binds much more strongly when cells enter S phase (Fig 3C), and to virtually all centromeres in HU-blocked cells (Fig EV2A). Rif1-ΔC594 on the other hand showed slightly higher association with centromeres during G1 phase than the full-length protein, but reduced association under conditions of HU blockage (Fig EV2A), suggesting that Rif1 association with centromeres is largely dependent on its C-terminus. This association is, however, unlikely to occur through Rap1 interaction, because previous genome-wide ChIP study did not find Rap1 associated with centromeres [33].

### Rif1 associates with replication origins

We observed binding of Rif1 and Rif1-ΔC594 to many replication origin sites genome-wide, with typical patterns observed illustrated

by Figs 2A, and 3A and B). Rif1 associated with both early-activated (e.g. *ARS607*, Fig 3A and *ARS1426*, Fig 3B left) and late-activated (e.g. *ARS1412*, Fig 3B right) origins, before and after origin initiation (e.g. at *ARS1412*, 90 min after release from α-factor). Rif1-ΔC594 was observed more frequently at origins than full-length Rif1 (e.g. at *ARS603* in Fig 2A), presumably reflecting increased availability of the truncated protein for binding to non-telomeric sites, caused by its release from telomeres.

Results of the ChIP-Seq analysis were confirmed at individual origins by ChIP followed by real-time quantitative PCR (ChIP-qPCR; Fig 4A). For example, using ChIP-qPCR we observed clear association of Rif1-ΔC594 with late origin *ARS1412* during G1 phase, association that was further increased at an HU block (Fig 4A right panel). The full-length Rif1 protein showed somewhat weaker association with *ARS1412*, again occurring in both G1 phase and HU-arrested cells (Fig 4A right panel). This association pattern is consistent with the ChIP-Seq result at the same locus (Fig 3B right). Rif1 and Rif1-ΔC594 showed similar binding patterns at an early origin (*ARS1426*, Fig 4A left), again consistent with the ChIP-Seq result (Fig 3B left). Although there was considerable scatter in the data (as is typical for ChIP-qPCR results close to the detection threshold), the results consistently revealed above-background binding and confirm that the ChIP-Seq profiles represent the genome-wide binding efficiencies of Rif1 and Rif1-ΔC594 reasonably well. These ChIP-qPCR analyses do also generally suggest higher binding levels of the Δ594 protein than wild type, possibly reflecting increased availability of the truncated protein due to its release from telomeres. By ChIP-Seq Rif1-ΔC594 also appears to show higher binding than full-length Rif1 at many loci (e.g. at *ARS1412*). However, peak heights in ChIP-Seq data may not provide an accurate measure of occupancy, due to limitations in the standardisation of ChIP-Seq results.

We performed peak-calling analysis on the ChIP-Seq data to allow comparison of the detected peaks with experimentally confirmed replication origins. Of the 410 replication origins that are experimentally confirmed in *S. cerevisiae*, we used a list of 329 origins that are not telomere proximal (> 15 kb from telomeres) and whose replication timing can be designated as either early or late (based on whether they have initiated in HU-arrested wild-type cells [34]). Within this list, 165 origins were assigned as early non-telomeric and 164 as late non-telomeric origins.

We observed full-length Rif1 associated with 104 of these 329 replication origins in G1 cells (Fig 4B; see also Fig EV2B); 47 of these were early and 57 late origins, indicating that Rif1 binds with no particular preference for early or late origins (Fig 4B). Rif1-ΔC594 associated with a larger number of origins in G1 phase, 174 of the total 329, but similar to the full-length protein showed no clear preference for either early or late.

We also observed clear binding of Rif1 proteins to origin sites in cells where replication was blocked by HU. In HU-blocked cells, full-length Rif1 bound to 111 of the origin sites; 86 of these corresponded to early and 25 to late-initiating origins, so that under HU blockage conditions full-length Rif1 shows a clear preference for early over late origin sites (Fig 4B). The highest number of origin sites was bound in Rif1-ΔC594 cells blocked with HU, where a large majority of both early and late origins (300 of the total 329) showed association with this truncated Rif1 protein. Overall, while Rif1 and Rif1-ΔC594 have somewhat different preferences for origin

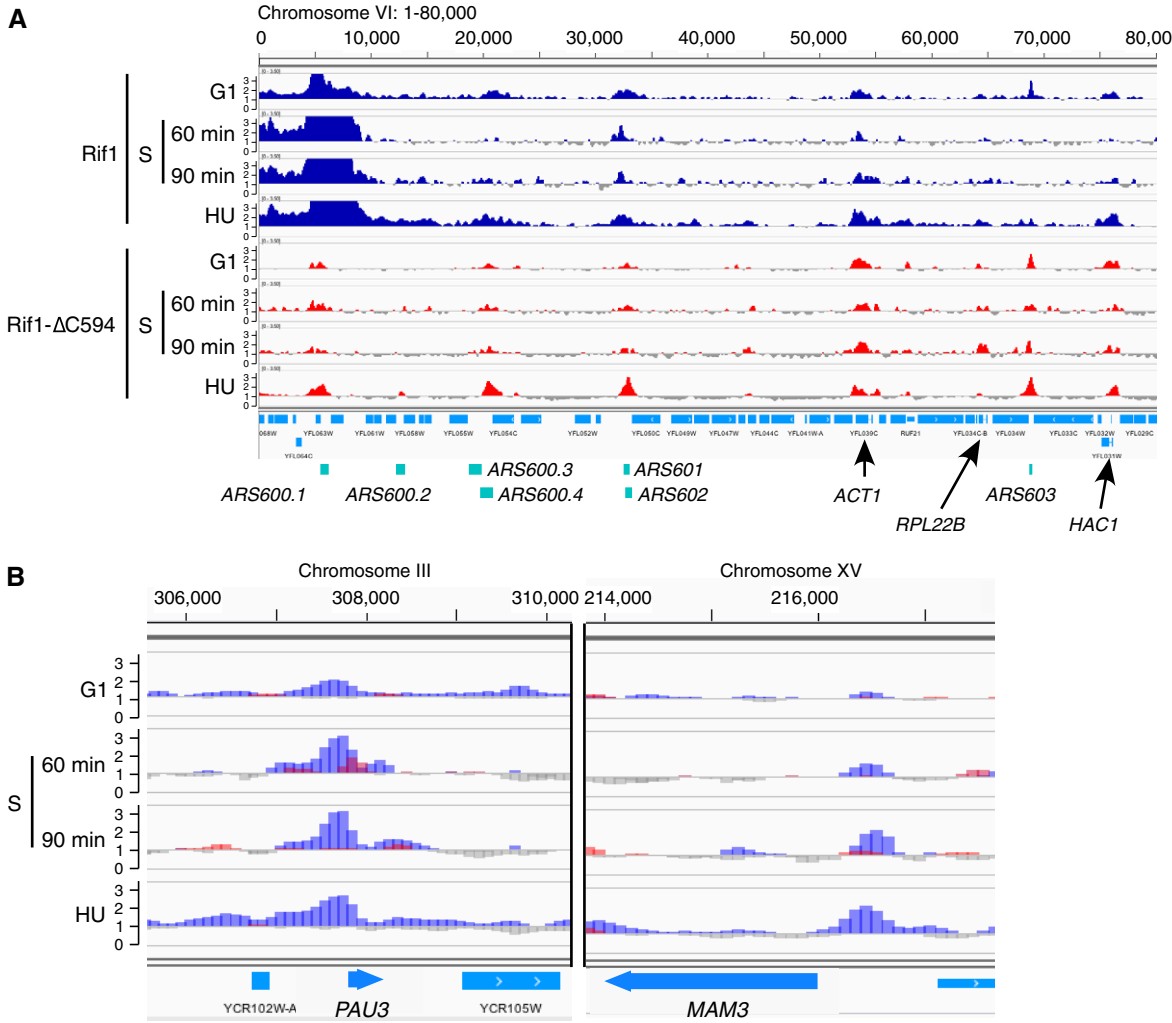

**Figure 2. Full-length and C-terminally truncated Rif1 proteins bind distinct chromosomal loci.**

A   Specimen overview of chromosome VI-left region showing results of ChIP-Seq analysis of Rif1 and Rif1-ΔC594 proteins. ChIP sequence reads were normalised against sequence reads from corresponding input samples, and relative enrichment is plotted for chromosome VI coordinates 1–80,000. Y-axis shows enrichment values (linear scale, range is 0–3.5). Values below 1 are shown in grey, and values above 1 (i.e. sequences enriched in ChIP samples) are coloured blue (Rif1) and red (Rif1-ΔC594). Plots show ChIP analysis results from cells arrested by α-factor (G1), released from α-factor at 16°C for 60 and 90 min, or released from α-factor into 0.2 M HU for 60 min at 23°C.

B   Rap1-dependent association of Rif1 with the promoter regions of Rap1-controlled genes. ChIP enrichment around *PAU3* (left) and *MAM3* (right), both genes whose transcription is controlled by Rap1. Values above 1 (i.e. enriched) shown by overlaid blue and red histograms for Rif1 and Rif1-ΔC594, respectively. Values below 1 shown in grey.

association, these preferences do not directly reflect the initiation time of origins, or their pre- or post-activation status.

**Replicating timing is maintained in *rif1-ΔC594* mutant**

The preference of full-length Rif1 for early over late origins in HU (Fig 4B) led us to consider the possibility that, after S phase begins, Rif1 can only bind origins that have already initiated. Such a possibility could be consistent with the binding of Rif1-ΔC594 to both early and late origin sites in HU, if it were the case that in the *rif1-ΔC594* mutant the replication timing programme was disrupted, so that almost all origins initiate before the HU block. To investigate

this possibility, we tested whether the replication timing programme is intact in the *rif1-ΔC594* strain by examining bromodeoxyuridine (BrdU) incorporation at early and late origins in HU-blocked cells. As assessed by BrdU incorporation, early origin *ARS607* was already activated in HU as expected (Fig 4C left). Two different late origins, *ARS422.5* and *ARS1412*, were inactive in both *RIF1+* and also in *rif1-ΔC594* strains (Fig 4C, middle and right), indicating that these origins remain inactive in HU and the replication timing programme is not lost in the *rif1-ΔC594* mutant. Both of these late origins showed somewhat increased BrdU incorporation in *rif1Δ* as expected based on previous analysis [11]. Overall therefore, the *rif1-ΔC594* mutant does not undergo wholesale disruption of the

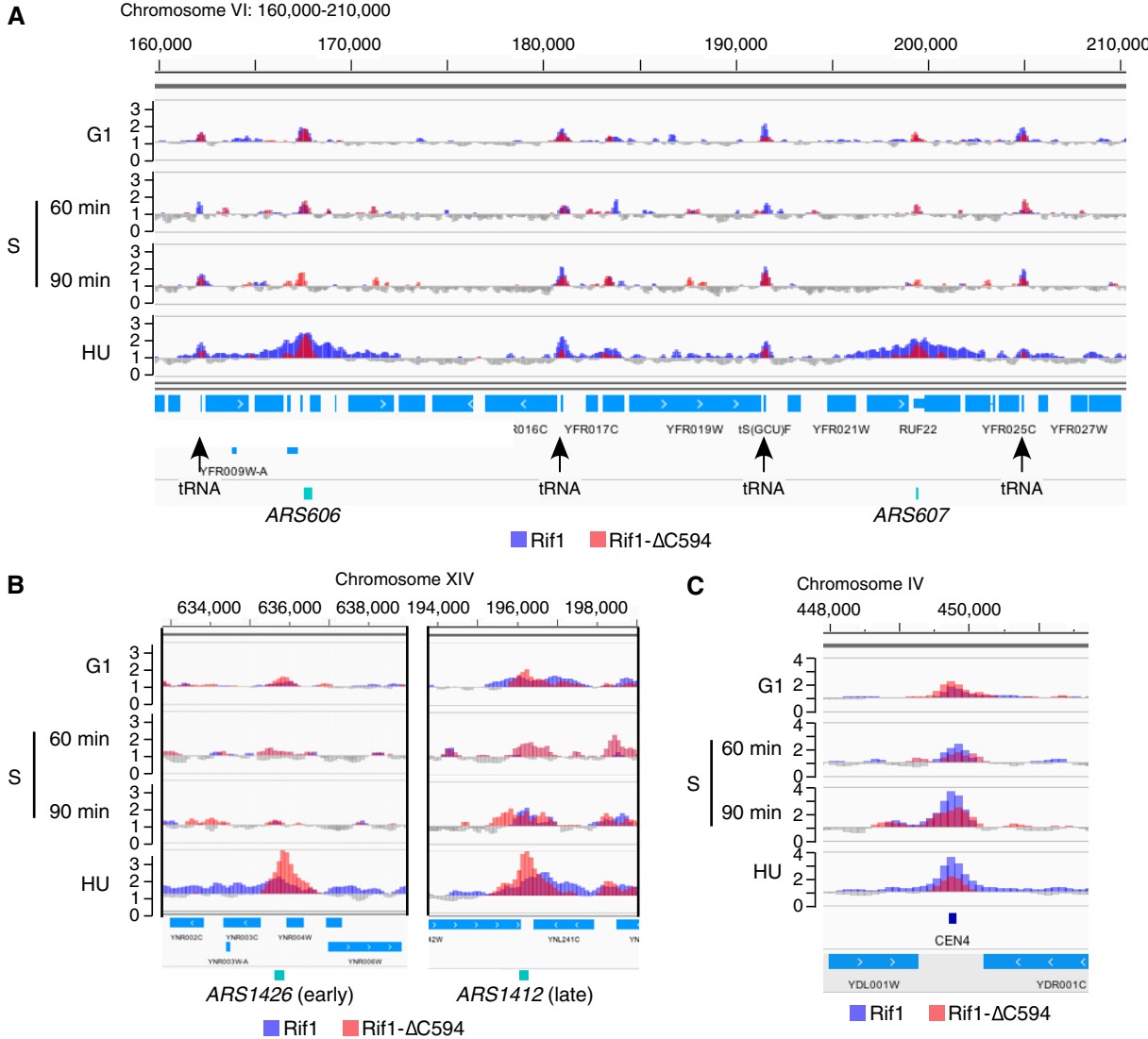

**Figure 3.  Rif1 and Rif1-ΔC bind replication origins, centromeres and tRNA genes.**

A   ChIP-Seq analysis of Rif1 and Rif1-ΔC594 proteins shows tRNA gene and origin binding, with widened peaks at early origins *ARS606* and *ARS607* in HU block. Plots show chromosome VI genome coordinates 160,000–210,000. Plot colours here and in following Figures are as in Fig 2B. Widened peaks are not observed in unperturbed S phase samples, or for Rif1-ΔC594.

B   Association of Rif1-ΔC594 at replication origins is enhanced in HU block. ChIP enrichment of Rif1 and Rif1-ΔC594 around early origin *ARS1426* (left) and late origin *ARS1412* (right).

C   Differential association of Rif1 and Rif1-ΔC594 to centromeres. ChIP enrichment of Rif1 and Rif1-ΔC594 around the *CEN4* locus.

replication timing programme but instead maintains the distinction between early and late origin activation. It moreover appears that Rif1-ΔC594 does associate with virtually all origins under HU-arrested condition (Figs 4B and EV2B), irrespective of whether origins have been activated or not.

**Rif1 protects nascent DNA at HU-blocked replication forks**

We noticed that peaks of full-length Rif1 at early origins tend to broaden in HU (e.g. *ARS606* & *ARS607*, Fig 3A). This broadened association seems to be specific to the HU-arrested condition, because it was not observed in the unperturbed S phase samples (Fig 3A).

Systematic analysis at early origins confirmed an increase in median peak width at early origins from 0.4 kb in G1 phase to 1.6 kb at the HU block (Figs 4D and EV3, heat maps). We did not observe such peak broadening at late origins (Figs 4D and EV3), suggesting that peak broadening requires origin activation, and probably therefore reflects association with replication forks stalled by HU inhibition. In a few cases (e.g. early origin *ARS1528*, Fig EV4A), we indeed observed peak splitting surrounding the origin site, consistent with the pattern representing association with blocked forks. Interestingly, the Rif1-ΔC594 mutant protein did not exhibit this pattern of replication fork association, as evidenced by the fact that peak broadening was not observed around early origins in HU (Fig EV3).

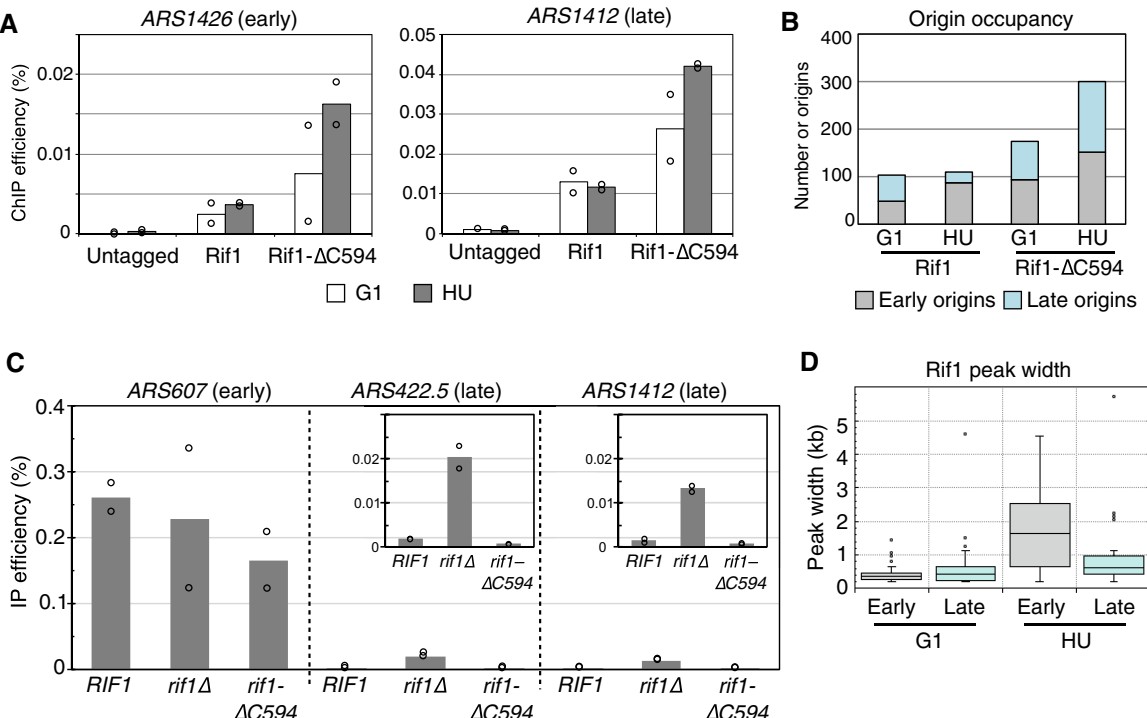

**Figure 4.  Analysis of origin binding reveals full-length Rif1 binds broad regions near early origins in HU.**

A  ChIP-qPCR confirmation of Rif1 and Rif1-ΔC594 association with replication origins. ChIP was performed using cells arrested in α-factor (open bars) or HU (grey bars), and qPCR analysis performed for early origin *ARS1426* (left) and late origin *ARS1412* (right). Values shown are "normalised ChIP efficiencies" obtained by subtracting the value obtained at a control locus (see Materials and Methods). Bars indicate the averages of two biological replicates, with values from each replicate shown by open circles.

B  Numbers of early and late origins associated with Rif1 and Rif1-ΔC594 peaks. Plot showing numbers of early and late origins bound by Rif1 and Rif1-ΔC594 in G1 phase and HU-blocked cells, based on peak-calling results.

C  Replication timing programme is intact in *rif1-ΔC594* cells. Replication of selected origins at an HU block analysed by BrdU incorporation. Cells were synchronised by α-factor and released into the medium containing 0.2 M HU and 1.13 mM BrdU. Plots show the percentage of total *ARS607*, *ARS422.5* or *ARS1412* DNA pulled down by IP with anti-BrdU, calculated from two biological replicate samples. Bars indicate the average of two biological replicates and open circles the values from each replicate. Insets in *ARS422,5* and *ARS1412* panels show the same data with Y-axis scales adjusted to 0–0.03%.

D  Peaks of Rif1 become broader in HU-arrested cells. Box and whisker plot shows peak widths of full-length Rif1 at early and late origins in G1 phase and HU-blocked cells. Analysis was performed on those origins detected by peak calling as associated with a Rif1 peak. Boxes show the range of 25th to 75th percentiles, with horizontal lines within the boxes representing 50th percentiles. Whiskers represent 95% confidence intervals. Outliers are presented as open circles. Numbers of origins analysed are as follows: 46 early origins in G1, 56 late origins in G1, 85 early origins in HU and 24 late origins in HU.

This pattern of Rif1 association suggested that yeast Rif1 might potentially play a role at blocked replication forks or on post-replicative chromatin. Since the Rif1-ΔC594 mutant does not show peak broadening, any such role might be expected to depend on the C-terminus of Rif1. Emerging data suggest that mammalian Rif1 stabilises nascent DNA at blocked replication forks [35]. We therefore tested whether yeast Rif1 protects nascent DNA at blocked replication forks, using DNA combing assays to analyse cells with nascent DNA labelled *in vivo* by iododeoxyuridine (IdU). Cells were released from α-factor in medium containing IdU. After 18 min, when cells have only just entered S phase so that only DNA synthesised from very early initiating origins will be labelled, IdU was removed and HU added to inhibit further replication. To examine the fate of the IdU-labelled nascent DNA, after either 1 or 1.5 h in HU genomic DNA was combed onto slides and the length of IdU-containing DNA tracts analysed (Fig 5A and B). At 0 h (the time of HU addition), all three strains showing IdU-labelled tract of median length around 18 kb, consistent with an early stage of S phase when

only some origins have initiated and bidirectional forks have travelled, on average, 9 kb each (Fig 5C). The length of nascent DNA tracts was similar in *RIF1+*, *rif1Δ* and *rif1-ΔC594* strains, indicating that in the three strains forks had progressed a similar distance from early origins. In *RIF1+* cells, the nascent DNA tract length remained stable throughout the subsequent 1.5-h incubation in HU. In *rif1Δ* cells, the nascent DNA tracts were in contrast noticeably eroded during the HU incubation period, with the median length decreasing from 21 to 13 kb in the first hour of the HU block. This result indicates that Rif1 is required to prevent degradation of newly-synthesised DNA at forks blocked by HU. In *rif1-ΔC594* cells, the IdU tract lengths were also shortened during the HU block when compared with *RIF1+* cells, consistent with the suggestion based on our ChIP results that the C-terminal region of Rif1 is important for protecting nascent DNA. The nascent tract shortening was not as severe in *rif1-ΔC594* as in *rif1Δ*, suggesting that while the protection of nascent DNA by Rif1-ΔC594 is significantly impaired, it may not be totally lost. Unexpectedly, we observed that despite the initial

shortening, after 1.5 h the median labelled tract length was increased in the *rif1-ΔC594* mutant (Fig 5C), an effect that was reproducible (Fig EV4B). While the reason for this observation is unclear, it could possibly reflect the complete loss of some particularly quickly degraded tracts in this *rif1-ΔC594* mutant context, if other tracts remain exempt from degradation.

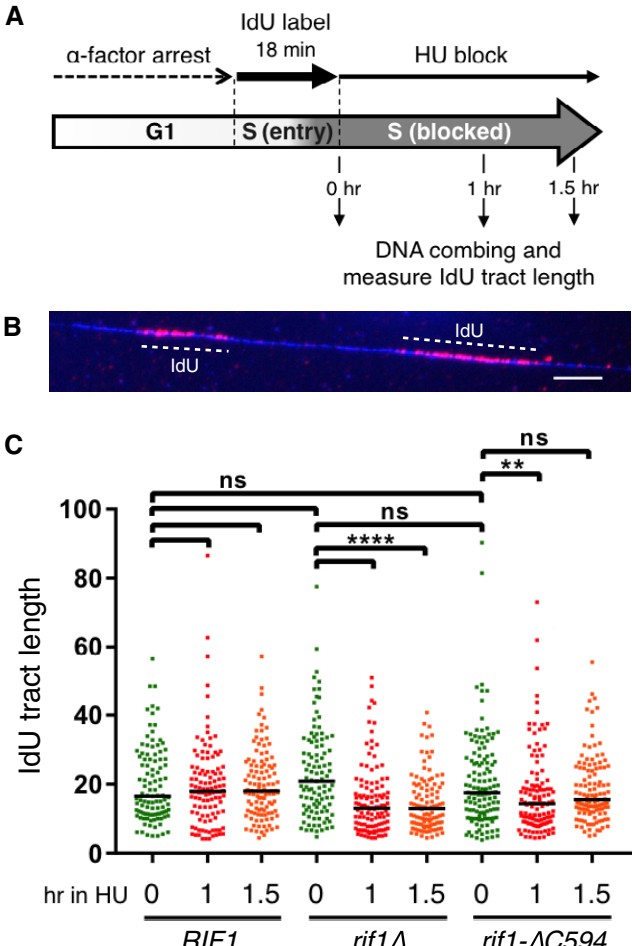

**Figure 5.  Rif1 protein is required to protect nascent DNA from degradation.**

A  Experimental scheme of nascent DNA protection assay. *RIF1, rif1Δ* and *rif1-ΔC594* strains (VGY85, CMY6 and CMY7 containing thymidine kinase gene insertions) were arrested with α-factor and released in the medium with 1.13 mM IdU. After 18 min of IdU labelling, IdU was removed by filtration and cells were resuspended in fresh medium with 0.2 M HU and 5 mM thymidine. This thymidine chase was included so that any residual fork extension or *do novo* origin activation occurring during the HU block would produce unlabelled DNA. After 0, 1 or 1.5 h, DNA combing was performed and IdU tract lengths analysed.

B  Specimen IdU tracts on a DNA fibre. DNA fibre is coloured blue and IdU tracts red. Scale bar is 10 μm (=20 kb).

C  Degradation of nascent DNA in the absence of Rif1. Plot shows distribution of IdU tract lengths obtained from DNA fibres prepared from cells incubated in the HU block for the time indicated. At least 100 tracts were measured for each condition. Black horizontal bars indicate median values. ** and **** indicate *P*-values less than 0.01 and 0.0001, respectively, obtained by Mann–Whitney–Wilcoxon test. ns means "not significant".

We obtained similar results in a second experiment, using a longer initial IdU labelling period of 22 min, which produced slightly longer initial tracts in the 0-h samples (Fig EV4B). Based on these results, we propose that the Rif1 protein is recruited to blocked replication forks, as indicated by our ChIP-Seq analysis, where it functions to stabilise nascent DNA and prevent its over-degradation.

## Rif1 associates with highly transcribed genes

As mentioned above, we found that both Rif1 and Rif1-ΔC594 tend to associate with the coding sequences of highly transcribed genes, such as *FBA1* on chromosome XI (Fig 6A, left panel). These sites often correspond to genes encoding ribosomal proteins (e.g. *RPL22B*, Fig 2A), other housekeeping genes (e.g. *ACT1*, Fig 2A), tRNA genes (Fig 3A) or genes involved in sugar metabolism (such as *FBA1* which encodes Fructose 1,6-bisphosphate aldolase required for glycolysis and gluconeogenesis). Consistent with their recruitment to highly transcribed genes, in α-factor-blocked cells Rif1 and Rif1-ΔC594 associate with *FIG1* gene, whose transcription is induced by α-factor [36]. This binding is lost once cells are released into S phase (Fig 6A middle panel). Conversely, Rif1 and Rif1-ΔC594 associate with the *RNR1* gene as cells enter S phase, and association is further increased in HU (Fig 6A right panel), mirroring the transcriptional control described for *RNR1* [37]. Note that Rif1 and Rif1-ΔC594 ChIP signals are significantly stronger than those obtained from an untagged control strain at the same sites (Fig 6A, lower two plots).

To assess whether Rif1 and Rif1-ΔC594 genuinely associate preferentially with highly transcribed genes, we identified those genes showing occupancy by Rif1 or Rif1-ΔC594 that extends across 90% of their coding sequences, and compared the expression levels of these genes with the expression levels of all *S. cerevisiae* genes using published results [38] (Fig 6B). Genes with high Rif1 or Rif1-ΔC594 occupancy showed a clear tendency to be highly transcribed (Fig 6B), with median levels of transcription fourfold to eightfold higher than the genomic average.

In budding yeast, the few genes with introns tend to be highly expressed [39], and genes showing high occupancy by Rif1 or Rif1-ΔC594 have higher then random likelihood of containing an intron (Table EV1). Genes encoding ribosomal proteins are generally highly expressed and showed a particularly interesting pattern of Rif1 association, with binding of Rif1-ΔC594 generally weakened in HU while that of full-length Rif1 was maintained (Fig EV5A *RPS31* plots and Fig EV5B heat maps). Rap1 regulates transcription of most ribosomal protein genes [40], and as expected, these genes frequently also show Rap1-dependent Rif1 binding of Rif1 in their upstream region (Fig EV5A and B). Another effect associated with strong transcription is the formation of R loops, DNA:RNA hybrid structures formed if a nascent transcript re-anneals to its template strand [41,42]. Many Rif1 and Rif1-ΔC594 peaks in coding sequences coincide with such R loop-forming loci ("DNA:RNA hybrid" track in Fig 6C and Table EV2).

Fission yeast Rif1 is suggested to bind G-quadruplex (G4)-forming sequences, so we also compared Rif1 and Rif1-ΔC594 ChIP patterns with positions of predicted *S. cerevisiae* G4-forming sequences [43,44]. However, at chromosome locations distant from telomeres, predicted G4 sites generally did not coincide with Rif1 and Rif1-ΔC594 peaks ("Predicted G4" track in Fig 6C and Table EV2).

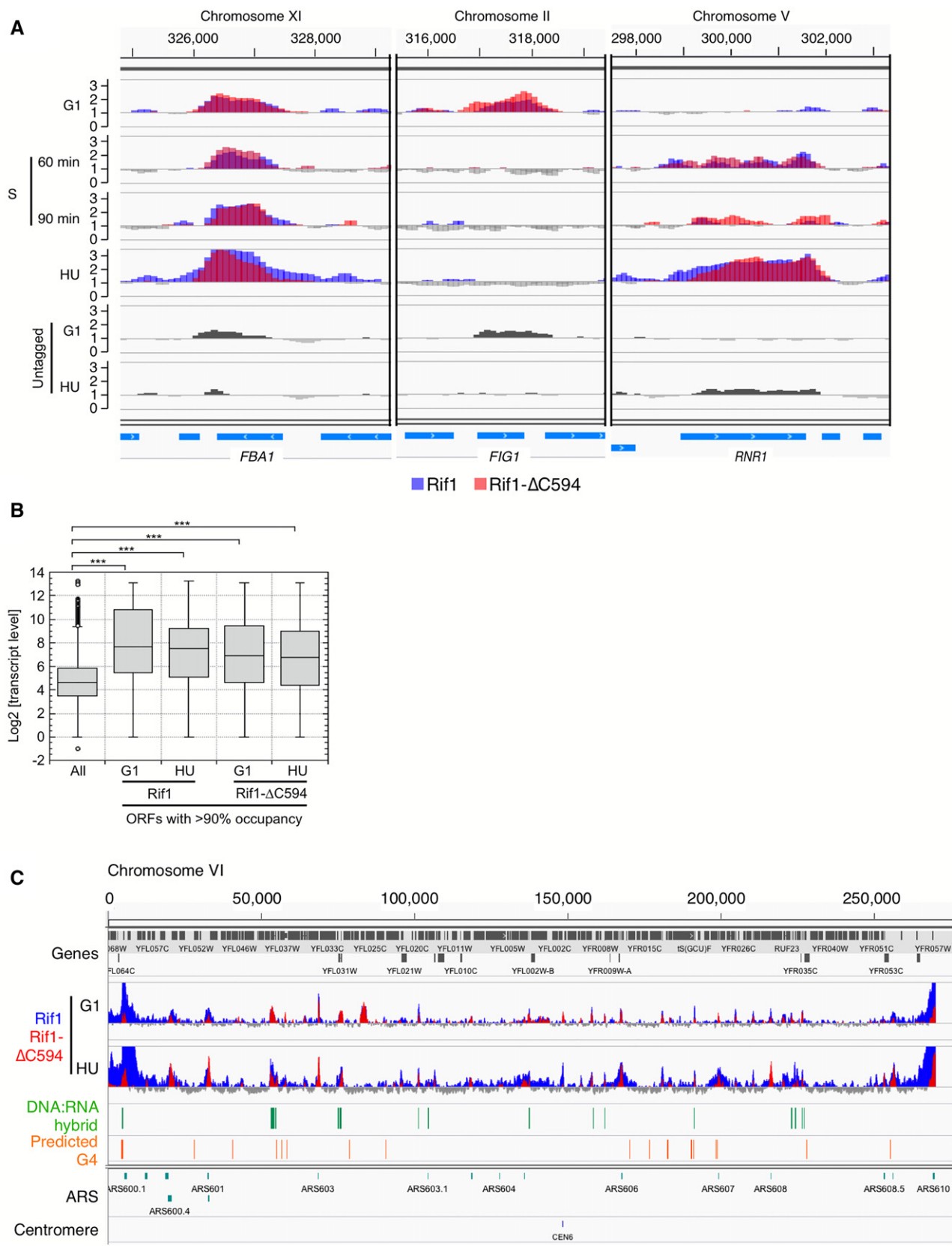

Figure 6.

**Figure 6. Rif1 and Rif1-ΔC564 associate with highly transcribed genes.**

A    Rif1 and Rif1-ΔC association with coding sequences of highly transcribed genes. Left panel shows association with the housekeeping gene *FBA1*, middle panel association with the mating pheromone-induced gene *FIG1* and right panel association with *RNR1*, which is expressed in S phase and induced further by hydroxyurea. Bottom two plots (dark grey) show results obtained at these loci using an "Untagged" control strain.

B    Rif1-associated genes tend to be highly transcribed. Genes showing occupancy by Rif1 or Rif1-ΔC594 that extends across 90% of their coding sequences were selected, and the transcription levels of these genes plotted compared to the transcription levels of all genes (shown at left). Number of genes analysed are 18 (Rif1, G1), 68 (Rif1, HU), 42 (Rif1-ΔC594, G1) and 64 (Rif-1ΔC594, HU). Boxes show the range of $25^{th}$ to $75^{th}$ percentiles, with horizontal lines within the boxes representing $50^{th}$ percentiles. Whiskers represent 95% confidence intervals. Outliers are presented as open circles. *** indicates that *P*-value obtained by Student's *t*-test was below 0.001.

C    Plot comparing chromosome-wide association of Rif1 and Rif1-ΔC594 with R loop-prone sites. Plot shows ChIP profiles of Rif1 and Rif1-ΔC594 across entire chromosome VI. R loop-prone sites are marked in green ("DNA:RNA hybrid" track in green), as previously assessed by S1-DRIP-Seq analysis. Also shown are positions of predicted G4-forming sequences ("Predicted G4" track in orange), and positions of replication origins (ARS) and centromere (CEN6).

## Discussion

We have examined genome-wide binding of the *S. cerevisiae* Rif1 protein, at several cell cycle stages. Compared to a previous, microarray-based analysis we identified numerous previously undescribed binding sites [3]. The possibility of identifying these sites was provided by the power of next-generation sequencing of ChIP samples, enabling analysis of millions of DNA fragments. This depth of analysis allows for a higher dynamic range within the data and therefore more effective identification of "secondary" binding sites, likely to be physiologically relevant but nonetheless obscured in previous studies by the very strong interaction of Rif1 with the telomeres, *MAT* locus and silent mating type cassettes. Thus, while our analysis effectively re-identified these strong, Rap1-dependent binding sites, it also permitted the identification of new types of Rif1 interaction with chromosomes. The use of a C-terminal truncation mutant, Rif1-ΔC594, allowed Rap1-dependent and Rap1-independent binding sites to be distinguished. We unexpectedly also found non-Rap1-associated sites that are bound by full-length Rif1 but not by Rif1-ΔC594, which are therefore likely to be under a control that requires the C-terminal region of Rif1. In this category are the associations with replication forks, centromeres (in S phase cells) and transcription units encoding ribosomal proteins (under HU-blocked conditions; Fig EV5).

An illustrative overview of the results is provided by Fig 6C, presenting data for the entire chromosome VI. Together with Fig EV1, Fig 6C illustrates all of the different binding site types we identified. Based on our analysis, we can categorise six different "types" of Rif1 chromosome binding site, as outlined:

(1) The highest levels of Rif1 binding are observed at telomeres, the *MAT* locus and silent mating type cassettes—all sites where Rif1 was already known to bind and function. Our observation that these sites are bound by Rif1 but not Rif1-ΔC594 confirms them to be Rap1-dependent sites of Rif1 binding.

(2) We also identified as Rif1 binding sites promoters that are regulated by Rap1. These sites were bound by full-length Rif1 but not Rif1-ΔC594, confirming the interaction to be Rap1-dependent. Although it might be expected that Rif1 would be present at sites where Rap1 acts as a transcriptional regulator, such sites had not been described before. Our observation of Rif1 at Rap1-regulated promoters implies that multiple copies of Rap1 are not needed for Rif1 recruitment. A previous genome-wide study indicated that that *RIF1* does not affect transcription of Rap1-controlled genes outside subtelomeric regions [45], suggesting that Rif1 recruitment to such promoters is not essential for transcriptional regulation by Rap1.

(3) Replication origins represent the third category of sites of Rif1 binding. Since both Rif1 and Rif1-ΔC594 bind to origins, this association is independent of Rap1. Since Rif1 is well established as a regulator of the replication timing programme in mammalian cells and the fission yeast *Schizosaccharomyces pombe*, we examined our results for any relation between Rif1 binding and origin initiation time. However, no such relationship was evident, except in the special case of telomere-proximal origins. The finding that Rif1 has no particular preference for either early or late-initiating origins is consistent with our knowledge that other than at telomeres, budding yeast Rif1 is not a major effector of the replication timing program. Instead, loss of Rif1 allows initiation from many origins, including normally early origins, under conditions of compromised DDK activity (i.e. in *cdc7-1* cells [9]), suggesting that Rif1 globally regulates origin activation rather than specifically suppressing late origins. Therefore, it is not surprising to find Rif1 localised to both early and late origins.

The mechanism through which Rif1 binds origins is unclear. While one possibility is that Rif1 interacts with a component of the pre-Replication Complex, this idea is not consistent with our observation of Rif1 binding to origin sites after their activation (e.g. at early origin *ARS607* in normal S phase samples, Fig 3A). Rif1 could potentially be recruited by ORC, which in yeast is believed to re-bind to origin sites quickly after their replication [46,47]. An alternative possibility is that Rif1 binds to origin DNA directly, perhaps mediated through the HEAT repeat domain which was recently identified as able to bind DNA directly [18].

(4) Our investigation has identified several types of genome interaction that were not predicted based on known *S. cerevisiae* Rif1 functions, the first of which is binding of full-length Rif1 in broadened peaks around early replication origins in HU-blocked cells. This association pattern could reflect Rif1 association during replication stress either with post-replicative chromatin or blocked forks. At one site, we observed peak splitting consistent with binding of bidirectional replication forks diverged from the replication origin itself. Since we did not observe Rif1 associated with replication forks in normal S phase, we suspect this pattern may reflect checkpoint-dependent recruitment of Rif1 specifically to blocked forks. Such checkpoint-dependent recruitment could potentially be controlled by the Rif1 C-terminal region, through a cluster of phosphorylation sites present in full-length Rif1 but absent in our Rif1-ΔC594 protein (Fig EV3) [25].

*Saccharomyces cerevisiae* Rif1 has not previously been identified interacting with replication forks, but mammalian Rif1 was already shown to be present at nascent chromatin [48], and emerging

results implicate mammalian Rif1 in protecting nascent DNA at stalled replication forks [35]. We tested nascent DNA protection in yeast using a DNA combing approach and found indeed that lack of *S. cerevisiae* Rif1 leaves nascent DNA exposed to abnormal degradation (Fig 5). It is unlikely that the short tracts observed after HU exposure are caused by new origin firing events labelled by residual IdU, since the appearance of shortened tracts was unaffected by the addition or omission of a thymidine chase after IdU labelling (not shown).

Defective protection of nascent DNA was also seen in the *rif1-ΔC594* strain, consistent with the defective recruitment of Rif1-ΔC594 to stalled forks, although in some experiments, the *rif1-ΔC594* fork protection defect was not as complete as in *rif1Δ* (compare Figs 5C with EV4B). One possibility is that Rif1-ΔC594 protein can partially protect nascent DNA, perhaps through passive or transient association with the forks. Increased availability of the truncated protein (since it is not sequestered at telomeres) may contribute to such a passive mechanism. Ongoing experiments will further test the molecular mechanism through which yeast Rif1 protects nascent DNA. It will be of particular interest to identify the nuclease responsible for nascent DNA degradation, and whether PP1 is required. It is unclear at this point how the nascent DNA degradation we observe may be related to previous studies in vertebrate cells implicating Rif1 in replication restart [49,50].

(5) A completely unexpected observation was binding of Rif1 to loci that are highly transcribed, including protein and RNA-encoding genes. Spurious binding caused by increased accessibility of highly transcribed loci (so-called hyper-ChIPability) is a recognised issue in ChIP analysis [51,52]. While we did observe some increased background signal at highly transcribed genes, binding of Rif1 to highly transcribed loci does not appear to be caused simply by "hyper-ChIPability", as in general signal clearly depended on the epitope tag (Fig 6A). Normalising the ChIP data with data from an untagged strain did not substantially change the Rif1 and Rif1-ΔC594 localisation patterns (data not shown). Our observation of distinct patterns for full-length Rif1 and the Rif1-ΔC594 mutant (e.g. at ribosomal protein genes Fig EV5) is also inconsistent with binding simply representing a consequence of increased locus accessibility, although we cannot exclude that "hyper-ChIPility" makes some contribution to the association pattern. At present, the functional significance of Rif1 binding to highly transcribed loci is unclear, but one intriguing possibility is that Rif1 recruitment is associated with the RNA:DNA hybrid "R loop" structures that tend to form at highly transcribed genes when an RNA transcript re-anneals to its template DNA. Interestingly, yeast Rif1 is reported to interact with the RNase H1 and RNase H2 enzymes that suppress excessive R loop formation (in cells over-expressing these RNase H proteins [53]). Another possibility is that Rif1 protects unwound DNA on the non-transcribed strand.

(6) Finally, we found that full-length Rif1 binds to centromeres (Figs 3C and EV2A). Interestingly, Rif1 associates more strongly with centromeres as cells traverse the cell cycle (Figs 3C and EV2A). It was recently shown that DDK-mediated phosphorylation of the kinetochore protein Ctf19 during G1 phase promotes recruitment of the cohesin loader complex to centromeres [54], raising the possibility that Rif1-PP1 regulates this process during the cell cycle.

The six different categories of binding site we have found together form a profile that differs significantly from the binding patterns described for Rif1 in other organisms. *S. pombe* Rif1 binds sequences with a tendency to form G4 DNA structures [55,56]. While we generally see no such tendency for *S. cerevisiae* Rif1 (Fig 6C and Table EV2), an intriguing possibility is that single stranded DNA exposed by transcription or impaired DNA replication may form G4 DNA that binds Rif1. In mouse embryonic stem cells (mESC), Rif1 has been described as occupying extended chromosomal domains, typically covering several megabases, which represent regions showing coordinated late replication timing [57]. The only chromosomal location where we see any similar pattern for *S. cerevisiae* Rif1 is close to telomeres (Figs 2A, 6C and EV1), where the full-length protein shows very high levels of binding extending over several kilobases, and where Rif1 is well-established as controlling replication timing over extended telomere-proximal domains [8–10,24,26]. Elsewhere in the genome the binding pattern of *S. cerevisiae* Rif1 is quite unlike that in mESCs and appears largely unrelated to the replication timing programme. Our discoveries are, however, consistent with the fact that *S. cerevisiae* Rif1 has fairly minor effects on replication timing at locations distant from telomeres. While mouse Rif1 was found at some transcription start sites, Foti *et al* did not find localisation of Rif1 to origin sites, centromeres, coding sequences of highly transcribed genes or replication forks (despite the fact that mammalian Rif1 has been detected as a nascent chromatin protein). The fact such binding sites were not identified in their study might mean that these modes of binding are specific to *S. cerevisiae* and are generally not conserved in mammalian cells, but equally might reflect the conditions or cell type used by Foti *et al* in their investigation, or be due to less intense binding sites being obscured by the extended domains of high Rif1 association.

A very recent report has described genome-wide chromosome association profiles of full-length and C-terminally truncated *S. cerevisiae* Rif1, obtained using the completely different methodology of ChEC-Seq (chromatin endogenous cleavage -Seq) [58]. Consistent with our findings, that study described strong Rif1 binding to telomeres and sub-telomeres through Rap1 interaction, and found association with internal replication origins that was enhanced by release of Rif1 from telomeres. Hafner *et al* do find Rif1 associating preferentially with origins whose timing it affects, but this observation may primarily reflect strong Rif1 binding in telomeric and telomere-proximal regions since origins close to telomeres were included in their assessment. The ChEC-Seq study did not report binding to the other sites (replication forks, centromeres, highly transcribed genes) that we have identified by ChIP-Seq. Direct comparison of the data-sets is complicated by the very different numbers of peaks identified in the two studies (~1,600 peaks here, compared to ~5,500 in the ChEC-Seq study).

To summarise, our investigation represents the first effective chromatin immunoprecipitation analysis of genome-wide binding sites of *S. cerevisiae* Rif1. It has identified several new modes of Rif1 genomic interaction, and in particular, it has led to our discovery that *S. cerevisiae* Rif1 protects nascent DNA at replication forks. Our approach, moreover, opens new avenues to understand how Rif1 is recruited to replication origins, blocked forks and sites of high transcription, which will enable substantial new insights into the molecular mechanisms deployed by this intriguing and multifunctional protein.

# Materials and Methods

### Yeast strains and plasmids

Yeast strains used in this study are in a W303 *RAD5*⁺ background and are described in Table 1. Strain CMY6 was created by replacing the *RIF1* gene of the strain VGY85 (Gali *et al* in preparation) with the *HIS3* gene using one-step PCR replacement. CMY7 was created by replacing the segment of *RIF1* encoding the C-terminal 594 amino acids with a 6His-3FLAG-*natMX* cassette, using the plasmid pSB54 as a template [59].

### ChIP-Seq analysis

Chromatin immunoprecipitation of Rif1-13myc (strain YSM20) and Rif1-ΔC594-13myc (strain KCY022) was performed essentially as described [60] using a monoclonal anti-Myc antibody [PL14] (MBL #M047-3) and Dynabeads Protein G (Dynal 10004D). Library DNA was prepared for Illumina sequencing using NEBNext Ultra II DNA Library Prep Kit for Illumina kit (NEB) following manufacturer's instructions and was analysed by Illumina HiSeq 2500. The

**Table 1.  Yeast strains used in this study.**

| Name | Relevant genotype | Background | Source/ reference |
|---|---|---|---|
| YK402 | *MATa bar1Δ::hisG ade2-1 can1-100 his3-11,15 leu2-3,112 trp1-1 ura3-1* | W303 *RAD5*⁺ | Hiraga *et al* [9] |
| YSM20 | *MATa bar1Δ ade2-1 can1-100 his3-11,15 leu2-3,112 trp1-1 ura3-1 Rif1-13Myc::HIS3MX6* | W303 *RAD5*⁺ | Sridhar *et al* [25] |
| KCY022 | *MATa bar1Δ::hphNT ade2-1 can1-100 his3-11,15 leu2,3-112 trp1-1 ura3-1 rif1-ΔC594-13Myc:: HIS3MX6* | W303 *RAD5*⁺ | This study |
| SHY538 | *MATa bar1Δ::hisG RAD5 ade2-1 his3-11,15 leu2-3,112 trp1-1 ura3-1 can1-100 cdc7-1* | W303 *RAD5*⁺ | Hiraga *et al* [9] |
| SHY614 | SHY538 *RIF11-6His-3FLAG:: kanMX* | W303 *RAD5*⁺ | This study |
| SHY616 | SHY538 *rif1-ΔC594-6His-3FLAG::kanMX* | W303 *RAD5*⁺ | This study |
| VGY85 | YK402 *trp1-1Δ::BrdU-InC-KanMX4* | W303 *RAD5*⁺ | Gali *et al* in preparation |
| CMY6 | VGY85 *rif1Δ::HIS3* | W303 *RAD5*⁺ | This study |
| CMY7 | VGY85 *rif1-ΔC-594-6His-3FLAG::natMX* | W303 *RAD5*⁺ | This study |

result was initially visualised and validated using DROMPA [61]. ChIP analysis from an untagged strain (YK402) was analysed similarly.

### Bioinformatic analysis of ChIP-Seq data

UCSC sacCer3 was used as the reference budding yeast genome throughout the study. Sequence reads from fastq files were mapped and sorted against this reference genome using Bowtie and SAMtools sort, respectively. ChIP peaks were detected using MACS2 callpeak. Rif1 ChIP peaks are a mixture of narrow peaks (e.g. peaks at *ARS* elements in G1 phase) and broad peaks (e.g. ORF peaks and peaks at early origins in HU arrest). Testing both "narrow" and "broad" peak options of MACS2, we opted to use "narrow peak" option, which often assigns multiple subpeaks in broad Rif1 ChIP peaks. Numbers of peaks detected are listed in Table EV3. Coverage of ORFs by ChIP peaks was analysed using BEDTools AnnotateBed. Average ChIP profiles and heat maps at centromeres, DNA replication origins and ribosomal protein genes were created by DeepTools computeMatrix and DeepTools plotHeatmap, using ChIP enrichment data (=ChIP data normalised by Input). Above procedures were performed using the Galaxy web interface (http://usegalaxy.org) [62]. ARS consensus sequence (ACS) position data were obtained from [63]. The list of yeast genes encoding ribosomal protein was obtained from the Ribosomal Protein Gene Database (http://ribosome.med.miyazaki-u.ac.jp/) [64], and their genome coordinates in 20110203 release (corresponding to sacCer3) were obtained from the Saccharomyces Genome Database (SGD; https://www.yeastgenome.org/). Positions of centromeres were also obtained from 20110203 release of SGD.

For illustration of ChIP results, enrichment of sequence reads in ChIP samples over corresponding input samples was calculated using DeepTools bamCompare, with bin size 100 bp and smoothing window 300 bp, and then visualised using Integrated Genome Browser (IGV) version 2.4.4. Genome coordinate information of known replication origins was obtained from OriDB (http://www.oridb.org) [27,28]. Positions of DNA:RNA hybrid detected by S1-DRIP-Seq in *rnh1Δ rnh201Δ* strain were obtained from [42]. Predicted positions of G4-forming sequence were obtained from [43]. Since genome coordinates of OriDB and G4 positions were based on an older genome assembly (sacCer1), the coordinates were converted to that of sacCer3 using LiftOver tool at UCSC Genome Browser (https://genome.ucsc.edu/cgi-bin/hgLiftOver). Information of budding yeast genes with experimentally identified and predicted introns was obtained from the Ares Lab Yeast Intron Database Version 4.3 UCSC (http://intron.ucsc.edu/yeast4.3/) [65]. Genome-wide transcript-level data were obtained from [38].

For comparison of the location of ChIP peaks and experimentally confirmed replication origins, the summit position of each ChIP peak was compared with the centre position of the closest confirmed *ARS* using a custom R script. If the distance to the closest *ARS* was less than 1 kb, the peak was marked to colocalise with the *ARS*. Where more than one peak was identified as colocalising with an *ARS*, ChIP profiles were inspected manually to determine where these corresponded to different subpeaks of a broad peak, or instead to two independent peaks (as in some cases, for example where one peak is close to an *ARS*, and a neighbouring peak is associated with a nearby genetic element such as a tRNA gene). When multiple

**Table 2.   List of primers for qPCR.**

| Locus | Orientation | Name | Sequence (5′ to 3′) |
|---|---|---|---|
| *ARS422.5* | Forward | CM24 | ACTGTCGGAATTGATGAGGGTG |
| | Reverse | CM25 | TCTCTTGCCTCCAAATTGTCCG |
| *ARS607* | Forward | VG54 | CGGCTCGTGCATTAAGCTTG |
| | Reverse | VG55 | TGCCGCACGCCAAACATTGC |
| *ARS1412* | Forward | VG64 | GCGTACGATGCGGTATGGAG |
| | Reverse | VG65 | TGCCGCACGCCAAACATTGC |
| *ARS1426* | Forward | SH709 | GCAAAGTCTTCCAAGAATCTGGTT |
| | Reverse | SH710 | GAGTTTCTATAGGTTTTAAAGGTGTGC |
| *IRS4* | Forward | SH713 | ACTCGGTTGTTGTTCATGTTGTC |
| | Reverse | SH714 | ATTTGGTAGTAAGCCCAAGCACT |

peaks were manually assigned as belonging to a single broad peak, the peak width was re-calculated for the merged peak. Note that peaks within 15 kb from each chromosome ends were excluded from this analysis, because the peaks tend to be fused with each other.

### ChIP-qPCR

ChIP of Rif1 and Rif1-ΔC594 was performed essentially as described [60] using antibody and beads as for the ChIP-Seq experiments. ChIP and corresponding input samples were analysed by LightCycler 480 II (Roche) using Light cycler SYBR Green master reagent (Roche). ChIP efficiency at each locus was calculated as the median of three technical replicates. "Normalised ChIP efficiency" was calculated by subtracting the ChIP efficiency value at the control (*IRS4* locus) at each strain and each culture condition from that of each locus tested. *IRS4* locus was chosen based on low association of Rif1 and Rif1-ΔC594 throughout the cell cycle in our ChIP-Seq data, as well as low background in "untagged" control experiments. See Table 2 for qPCR primers used.

### BrdU-IP-qPCR

BrdU-IP was performed essentially as described [66]. Strains containing thymidine kinase insertion constructs, VGY85, CMY6 and CMY7, were arrested with α-factor and released into fresh media containing 1.13 mM (400 μg/ml) BrdU and 0.2 M HU, and cultivated for 60 min. Genomic DNA was isolated as previously described [67]. 1 μg of total genomic DNA was immunoprecipitated with 10 μg of anti-BrdU antibody (ab2285, Abcam). BrdU-labelled DNA was then extracted using Dynabeads Protein G (Dynal) and purified using QIAquick PCR Purification Kit (QIAGEN). qPCR analysis was performed as above, and IP efficiency calculated as the percentage of total sequence pulled down. See Table 2 for qPCR primers used.

### DNA combing

Cells were arrested with α-factor, released into fresh media containing 1.13 mM IdU and cultivated for 18 min (for the experiments shown in Fig 5) or 22 min (Fig EV4B) at 30°C. Cells were then filtered, washed and resuspended in fresh media containing 0.2 M

HU. Note that 5 mM thymidine was also included during the HU incubation to minimise the labelling of any ongoing DNA synthesis by residual IdU. Cells were collected at 0, 1 and 1.5 h (2 h for Fig EV4B) and encapsulated in plugs of low melting temperature agarose. Spheroplasting was carried out in agarose plugs, followed by genomic DNA preparation using FiberPrep DNA extraction kit (Genomic Vision) according to manufacturer's instructions. DNA combing was performed using FiberComb instrument (Genomic Vision). Coverslips with combed DNA were processed for immunostaining with anti-IdU (Becton Dickinson 347580) and anti-ssDNA (Millipore MAB3034) followed by appropriate secondary antibodies with fluorescent conjugates. IdU tracks were imaged under a Zeiss Axio Imager.M2 microscope equipped with Zeiss MRm digital camera, with a Zeiss Plan-Apochromat 63×/1.40 Oil objective lens. Images were analysed using ImageJ software. IdU-labelled tract lengths were measured, requiring that tracts must be at least 2 μm in length, separated from each other by 5 μm or more, and lie on a fragment at least 50 μm in length with the tract finishing at least 5 μm from the fragment end as visualised by ssDNA antibody. Length of the IdU tracks (in μm) was converted to kilobases using the predetermined value (2 kb/μm) for the DNA combing method.

## Data availability

ChIP-Seq data and corresponding input data were submitted to ArrayExpress under accession number E-MTAB-6736.

**Expanded View** for this article is available online.

### Acknowledgements

We thank Javier Garzon and Vamsi Krishna Gali for discussion and advice on methods, and Takashi Kubota for helpful comments on the manuscript. This work was supported by Cancer Research UK Programme Award A19059 to ADD and SH. KS was supported by Grant-in-Aid for Scientific Research on Priority Areas (15H05970 and 15K21761) from Ministry of Education, Culture, Sports, Science and Technology, Japan.

### Author contributions

SH, KS and ADD conceived the research project. SH, CM, YK and KRMC performed the experiments. SH, CM, YK and SS analysed the data. SH and ADD wrote the manuscript.

## Conflict of interest

The authors declare that they have no conflict of interest.

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
