## [Review Process File · EMBO Reports]

Budding yeast Rif1 binds to replication origins and protects DNA at blocked replication forks

Shin-ichiro Hiraga, Chandre Monerawela, Yuki Katou, Sophie Shaw, Kate R.M. Clark, Katsuhiko Shirahige, Anne D. Donaldson

Review timeline:	Submission date:	11th Apr 18
	Editorial Decision:	7th May 18
	Revision received:	27th Jun 18
	Editorial Decision:	11th Jul 18
	Revision received:	17th Jul 18
	Accepted:	23rd Jul 18

Editor: Esther Schnapp

Transaction Report:

1st Editorial Decision

7th May 18

Thank you for your patience while your manuscript was peer-reviewed at EMBO reports. We have now received the full set of referee reports that is pasted below.

As you will see, all referees acknowledge that the findings are interesting and that they should be published. They only have minor concerns that should all be addressed. Please do discuss your data in light of the recent Cell reports paper, and address the concern by both referees 1 and 2 regarding the role of the rif1 mutant in protecting nascent DNA. It will also be very important to provide all statistical information in the figure legends.

Given these constructive comments, we would like to invite you to revise your manuscript with the understanding that the referee concerns must be fully addressed and their suggestions taken on board. Please address all referee concerns in a complete point-by-point response. Acceptance of the manuscript will depend on a positive outcome of a short, second round of review. It is EMBO reports policy to allow a single round of revision only and acceptance or rejection of the manuscript will therefore depend on the completeness of your responses included in the next, final version of the manuscript.

Given the competing Cell reports paper, I suggest that you submit the revised manuscripts as soon as possible, preferably within the next 2 weeks. Given the 6 main figures, I suggest that we publish the manuscript as a full article, as it is laid out now.

REFeree COMMENTS

Referee #1:

Rif1 is a multifunctional protein, many functions of which are conserved from yeast to humans,

including the ability to delay replication of nearby origins. ChIP has been challenging in budding yeast due to the enormous amount of Rif1 at telomeres, which masks its presence at other sites using ChIP to microarrays. Hence, Rif1's role in replication timing of budding yeast origins has been limited to studies of telomeric origins. Hiraga et al., set out to tackle this challenge by removing the C terminal section of the Rif1 protein, which targets it to telomeres, and then performing the more sensitive ChIP-seq, allowing the authors to map Rif1 chromatin interactions throughout the genome. The authors first confirm the replication functionality of the Rif1 truncation by showing it can still repress growth of *cdc7-1* mutations similar to wild type Rif1. ChIP-seq was then performed on both wild type and truncated Rif1 at different times during the cell cycle (G1, early S, and late S) and under HU stress. As expected, the wild type Rif1 protein bound highly at the telomeres while the truncated Rif1 did not bind telomeres. Since recruitment of Rif1 to telomeres occurs through an interaction of its C-terminus with Rap1, this experiment revealed Rap1 dependent and independent Rif1 chromatin-binding sites. Unexpectedly, some of the Rap1 independent Rif1 binding sites corresponded to highly transcribed genes and the cell cycle regulated binding to these genes was coincident with up-regulation of their expression. Rif1 was also found to bind to early replicating centromeres in a Rap1 independent manner. Both wild type and truncated Rif1 was localized to internal replication origins independent of their replication timing, suggesting that Rif1 has a minimal role in regulating replication timing at non-telomeric origins in budding yeast. Another unexpected finding the authors noticed was a broadening of wild type Rif1 peaks at early origins in HU treated cells. Peak splitting was also observed around early origins in HU treated wild type Rif1. Peak broadening and peak splitting were not observed in truncated Rif1 ChIP. Peak splitting suggests that wild type Rif1 is at the sites of stalled replication forks while the mutant is not, so the authors looked to see whether Rif1 was acting to protect nascent DNA at stalled forks by DNA fiber analysis. The authors found a shortening of replication tracks in Rif1 knockout cells, which they interpret as evidence for Rif1's ability to prevent resection of nascent DNA. Overall, this paper provides convincing evidence for Rif1's Rap1 independent functions in budding yeast. The authors overcome a major gap in the field by mapping Rif1 binding sites. The paper reports several very surprising results that will warrant re-interpretation of the literature and are important to report. I think, given the surprise that Rif1 is not enriched at late origins, the authors should tell us whether the cell cycle regulated binding of Rif1 is different at early vs. late origins or not.

The conclusion that Rif1 plays a role in protecting stalled replication forks from degradation was less convincing. Their data are consistent with Rif1 being necessary for elongation of forks near HU-origins. However, what is happening to the map positions of the HU replication tracks on either side of HU origins in Rif1 mutant cells (BrdU-seq)? Also, why did the authors map track lengths (fibers) in Rif1 null cells when the entire paper is comparing WT to truncation mutant? Most important, however, is the leap of logic from shorter replication fork tracks to the conclusion that Rif1 is preventing the degradation of recently synthesized DNA. Maybe Rif1 promotes replication restart, fork elongation rates - it seems that there are other interpretations that would require further investigation to sort out. I would recommend that the authors focus their conclusions on their intriguing findings that are clear and tone down the conclusions regarding resection and fork degradation.

Other small comments:

- Figure 2B and 4E could use at least one more example.
- Is there a difference between the fibers in Figure 5B?
- Truncated Rif1 binding is strikingly higher than wild type Rif1 at GAL4 upon induction, discussion of this result is lacking.
- The authors conclude that there is 'a large portion of R-loops' overlapping with Rif1 binding sites but there is no evidence shown. Quantification of the overlap between Rif1 binding sites and R-loops/G4 motifs is lacking.

Referee #2:

The Rif1 protein has emerged in several recent studies as a multifunctional, global regulator of genome replication and stability, in large part through its recently discovered role as a co-factor for

the conserved protein phosphatase PP1. One of the roles for the Rif1-PP1 complex, conserved from yeast to humans, is to counteract potentially "premature" phosphorylation of the MCM helicase complex by S-phase activator DDK, and thus presumably premature origin activation. In addition, in yeast, Rif1 is also a Rap1 interacting protein and via this role contributes to telomere length control and transcriptional silencing.

This study begins with a high-resolution ChIP-Seq analyses of Rif1 binding in yeast, and succeeds in defining many new Rif1 target regions across the chromosome worth deeper mechanistic consideration in future studies. This represents the first high-resolution analyses of Rif1-chromosome target site binding on a genome-scale. Given Rif1's central importance in genome stability, these data are important to the field and of high potential physiological relevance. Thus this basic result, along with revealing a role for Rif1 at stalled replication forks (RF) in protecting nascent DNA, fits the stated mission of the EMBO Reports journal well: describing an interesting set of results of high potential physiological relevance without delving into the mechanisms underlying the outcomes.

There are some concerns about presentation/logic that I think could be dealt with by the authors with some editing. It's mostly about how some of the writing leads to confusion for the reader:

1. From the chromosome VI scan in Figure 1, excluding the telomeric regions, Rif1 and rif1DC594, a mutant version of Rif1 that prevents Rif1 from binding to Rap1, actually have very similar binding profiles across this chromosome don't they?. That is OK (and, I think, good news, actually)--these are the data, but this general observation does not mesh well with the logic of using the rif1DC594 mutant in the first place as it was presented. For example, line 28: "release of Rif1 from telomeres increases Rif1 binding to other genomic sites, revealing 100's of Rap1-dependent and Rap1-independent Rif1 sites" ...This conclusion is not supported by the data in Figure 1, and so was quite confusing as a background rationale. In terms of overall distribution of Rif1 target sites in Figure 1A, it simply appears that rif1DC594 produced a few weaker peaks compared to wild type Rif1, while it does not appear that many new sites were identified by using this mutant. While it is fair to conclude that you have identified many new Rif1 chromosomal target sites, I am not sure I am convinced that you reached that success because you used a rif1DC594 mutant. I think it is the ChIP_Seq technology. As I said, these are the data, but one reason I in some ways prefer this outcome is that I am not sure how interesting it would be to find target sites that only a rif1DC594 mutant could bind...
2. Along these same lines, and related to the larger issue of determining the "no binding" chromosomal baseline for such experiments, the authors state: line 96: Rif1's strong association with Rap1 "tends to obscure Rif1 binding to other loci"but that is simply not evident in the data--I see a lot of new "non-tel" sites for wild type Rif1 (blue) that do not seem to be obscured by Rif1's ability to bind Rap1 (because I see most of the same sites in rif1DC594 as well). The mutant and the wild type Rif1 have similar profiles. So from the data in Figure 2 it does not look as if releasing Rif1 from telomeres by inactivating its ability to bind Rap1 had all that much of an effect, other than reducing Rif1's association relative to the mutant at Rap1 sites (telomeres, ribosomal protein genes). How exactly were the binding data normalized in WT compared to the mutant? How is the baseline (no binding) determined? Presumably the chromosomal baseline for WT, because so much binding occurs at telomeres, would be higher than that for the mutant and that could make it appear that the rif1DC594 binds some regions more effectively even if it does not--has a "correction" for that been made for the baselines in Figure 2 in WT vs mutant? If you generate baselines for both WT and mutant Rif1 where the telomeric regions (super strong Rif1-binding) are removed, does it now appear that WT Rif1 actually binds a lot better to more target sites than the mutant? How many sites showed greater association in WT Rif1 versus rif1DC594 and vice-versa? It seems like strong conclusions were being made without data to support them in this instance.
3. Rif1 binding to replication origins was clear and an important new piece of data. The differences between the behavior of Rif1 versus the rif1DC594 mutant at origins in cells arrested in HU was very interesting. The ability for Rif1 to interact with Rap1 is not needed for Rif1 to get to origins, but that C-terminal region that is nevertheless needed for the extended distribution of Rif1 around origins that are arrested. Interestingly though there was no effect on replication origin function as measured by BrdU-IP between WT and rif1DC594 (Fig 4C)--Are blue and orange bars simply biological/independent replicates? I assume from the Figure legend yes, but why are they colored differently if there is no key in the figure indicating why? And if these represent replicates, why are they so different for rif1D at ARS607? It seems that the major result is how differently WT and the

DC594 mutants act at origins arrested in HU and that point gets a little lost in some of the other details that seem less impactful.

4. The authors use DNA combing to test whether the broad association of Rif1 at origins arrested in HU that they observed might be relevant to stabilizing nascent DNA at stalled RFs, a phenotype that is emerging as relevant to Rif1 function in mammalian cells and so potentially extremely exciting. They found that indeed in *rif1* Δ C594 cells nascent DNA was degraded suggesting that Rif1 in yeast functions to stabilize nascent DNA at stalled RFs. The issue that surprised me is that they didn't test what the *rif1* Δ C594 did in this assay and because they have the mutant and have been using it as a kind of control throughout the study this omission seemed odd, even though EMBO Reports does not require mechanism. Given that *rif1* Δ C594 does not show the same spread-out binding distribution as Rif1 at HU arrested forks but still binds to origins, this seemed like an obvious way to test whether that binding pattern of WT Rif1 was indeed relevant to nascent DNA protection.

5. I was not convinced that the binding patterns of Rif1 over highly transcribed genes were not related to "hypechippability" artifacts based on the data shown even if the patterns differ between Rif1 versus *rif1* Δ C594--The no-tag control data that they report that they generated (as data not shown) should also be shown in Fig6. What kind of overlap did you observe between the ORFs you defined in this way and those discussed in the two relevant ChIP_Seq-Bias studies? Understanding how these types of signals can be generated, and their potential variation from experiment to experiment is worth noting, regardless. In addition, there is an additional relevant reference besides that of Teytelman (Park, Lee, Bhupindersingh, Iyer, 2013, PLoS One)

Referee #3:

In their manuscript 'Budding yeast Rif1 binds to replication origins and blocked replication forks to protect nascent DNA' Hiraga et al. analysed the genome-wide binding sites of the budding yeast Rif1 protein, a factor involved in replication timing and telomere length control. Using a C-terminal deletion of Rif1 enabled them to investigate its role in a Rap1 independent manner.

In particular, Hiraga et al. show a most complete set of DNA associations in *S. cerevisiae* for RIF1, *rif1* and *rif1* Δ C594 and identified that Rif1 can bind a lot of genomic regions independent of Rap1 when deleted for aa594 onward. Interestingly, Rif1 and *rif1* Δ C594 have different preferences for origin binding, which doesn't correlate with replication status/timing and no replication timing defect was observed for *rif1* Δ C594. When treated in HU, deletion of Rif1 results in shorter nascent DNA tracks, suggesting a protective mechanism by Rif1. Finally, the authors found that Rif1 Δ C594 is recruited to highly transcribed class II and III genes.

Although some studies already identified the major binding sites of Rif1, this manuscript represents the most detailed description of the genomic binding sites of Rif1 (Rap1- dependent and - independent) under different growth conditions up to date. This manuscript provides a description of many new genomic Rif1 binding events and discusses their biological implication. Future work, as part of independent studies, will provide more evidence for each of the 6 classes of binding sites, but are not necessary for publication of this manuscript.

Overall, the paper makes a very nice contribution towards the field and raises new hypotheses of Rif1 function in *S. cerevisiae* and beyond.

Major comments:

I. We suggest to add the data of Hapfner et al., 2018, Cell Reports, which corroborates the findings of the authors. This gives the authors an opportunity to discuss similarities in the context of their 6 class model and differences they observed due to different techniques and mutants.

II. ChIP-qPCR: Please comment on how many biological replicates used in all relevant figures, why was PAC2 used as control gene? Its centromeric location (<15kb) could make it a suboptimal candidate gene, maybe the authors should use alternative control genes? The given values should be rather described as average of biological replicates with standard deviation instead of median. Better information on error bar use throughout the manuscript. Fold enrichments below 1 may indicate unspecific enrichment of DNA (high background?)

Minor comments:

I. A graphical depiction of the 6 class model would be helpful for understanding.

II. Line 127: Figure legend and text appear not to fit in respect of permissive/semi-

permissive/restrictive labelling or the text is unclear.

III. FigS2/S5/S6: Colour code is misleading: Red usually implies high occupancies/ binding strength

IV. Fig.3C add labelling below as in B.

V. Fig.4A: show more than one locus, early vs. late origin would be helpful

VI. Fig.4B: use a stacked column layout to make the differences in early vs. late origins more obvious

VII. Fig.4B lacks a title

VIII. Fig. 4C lacks labelling of the orange and blue columns - it is unclear what is shown.

IX. Fig.4D lacks a title and proper labelling

X. Fig.5B: poor labelling, no title, maybe better micrographs available? Why is Rif1- Δ C594 not included?

XI. Fig.S3C: the presented recruitment is not convincing, as recruitments are measured within the background

XII. Fig.S4: same as in comment VI

XIII. Fig.6C: change y-axis labelling according to e.g. Fig.4A, add error bar for 'untagged +GAL' sample. How many biological replicates are shown?

XIV. ChIP-qPCR primers and corresponding sequences are not included

XV. Fig.S6: Rif1- Δ C594 lacks promoter associated peak, is this due to the inability to bind to Rap1? How does Rap1 behave at sites of high transcription?

XVI. No p-values (significances as indicated by different amounts of *) explained in the manuscript as well as no indication on error bar usage throughout the manuscript.

We thank the Reviewers for their interest in the work and their useful comments, which we address below. All Figure and page numbers refer to the revised manuscript, unless otherwise mentioned.

Referee #1:

Rif1 is a multifunctional protein, many functions of which are conserved from yeast to humans, including the ability to delay replication of nearby origins. ChIP has been challenging in budding yeast due to the enormous amount of Rif1 at telomeres, which masks its presence at other sites using ChIP to microarrays. Hence, Rif1's role in replication timing of budding yeast origins has been limited to studies of telomeric origins. Hiraga et al., set out to tackle this challenge by removing the C terminal section of the Rif1 protein, which targets it to telomeres, and then performing the more sensitive ChIP-seq, allowing the authors to map Rif1 chromatin interactions throughout the genome. The authors first confirm the replication functionality of the Rif1 truncation by showing it can still repress growth of *cdc7-1* mutations similar to wild type Rif1. ChIP-seq was then performed on both wild type and truncated Rif1 at different times during the cell cycle (G1, early S, and late S) and under HU stress. As expected, the wild type Rif1 protein bound highly at the telomeres while the truncated Rif1 did not bind telomeres. Since recruitment of Rif1 to telomeres occurs through an interaction of its C-terminus with Rap1, this experiment revealed Rap1 dependent and independent Rif1 chromatin-binding sites. Unexpectedly, some of the Rap1 independent Rif1 binding sites corresponded to highly transcribed genes and the cell cycle regulated binding to these genes was coincident with up-regulation of their expression. Rif1 was also found to bind to early replicating centromeres in a Rap1 independent manner. Both wild type and truncated Rif1 was localized to internal replication origins independent of their replication timing, suggesting that Rif1 has a minimal role in regulating replication timing at non-telomeric origins in budding yeast. Another unexpected finding the authors noticed was a broadening of wild type Rif1 peaks at early origins in HU treated cells. Peak splitting was also observed around early origins in HU treated wild type Rif1. Peak broadening and peak splitting were not observed in truncated Rif1 ChIP. Peak splitting suggests that wild type Rif1 is at the sites of stalled replication forks while the mutant is not, so the authors looked to see whether Rif1 was acting to protect nascent DNA at stalled forks by DNA fiber analysis. The authors found a shortening of replication tracks in Rif1 knockout cells, which they interpret as evidence for Rif1's ability to prevent resection of nascent DNA.

Overall, this paper provides convincing evidence for Rif1's Rap1 independent functions in budding yeast. The authors overcome a major gap in the field by mapping Rif1 binding sites. The paper reports several very surprising results that will warrant re-interpretation of the literature and are important to report. I think, given the surprise that Rif1 is not enriched at late origins, the authors should tell us whether the cell cycle regulated binding of Rif1 is different at early vs. late origins or not.

This analysis is contained within Fig. 4B of the revised version and described on page 7-8 of the revised

text. The truncated Rif1- Δ C594 protein binds 174 of the possible total 329 origins in G1 phase, with no particular preference for early or late origins. In HU the number of origins bound by Rif1- Δ C594 increases to 300, still with no clear preference for early or late (and therefore binding occurs apparently irrespective of whether the origin has or has not been activated). Replication timing is

intact in this mutant (Fig. 4) so there appears to be no straightforward relationship between whether a replication origin is Rif1-associated and its initiation time.

Full-length Rif1 protein binds 104 of the possible total 329 origins in G1 phase, again with no particular preference for early or late origins. In HU the number of origins bound by full-length Rif1 increases to 111. 86 of these are early-replicating sites. However, the obvious interpretation that in HU Rif1 shows a preference for early origins is here complicated by the fact that the generally broad peaks observed means we cannot distinguish 'origin' from 'blocked fork' binding. What is clear is that in this condition fewer late origins are bound by Rif1. We suspect the depletion from late origins may reflect that most of the 'available' pool of Rif1 in the nucleus gets recruited to blocked forks.

The conclusion that Rif1 plays a role in protecting stalled replication forks from degradation was less convincing. Their data are consistent with Rif1 being necessary for elongation of forks near HU-origins. We don't see how our data can be interpreted as consistent with 'Rif1 being necessary for elongation of forks...'. Before HU addition fork elongation appears similar in the wild-type and $\Delta rif1$ strains (Fig. 5 and Fig. EV4), since the pre-labelled regions start out at very similar length in the wild-type and $\Delta rif1$ strains when HU is added (at 0 hr), with the labelled DNA then either remaining intact (wt) or getting removed ($\Delta rif1$ mutant) in the HU block. Does the reviewer mean that we could be measuring some amount of ongoing replication occurring during the HU block? Such replication is unlikely to be visualised by our assay, since we add a high concentration thymidine chase along with the HU at the time the IdU is removed. So while indeed there could be restart or elongation defects in the $\Delta rif1$ mutant, in our experimental design such effects would not produce IdU-labelled DNA, since no label is present in the 'restart' phase. And indeed very little if any elongation is apparent in our wild-type strain during the HU block, as designed by our experimental procedure.

We have clarified our explanation of the experimental design to more fully explain our reasoning (see page 10).

However, what is happening to the map positions of the HU replication tracks on either side of HU origins in Rif1 mutant cells (BrdU-seq)?

We do not have BrdU-seq data for these experiments. While such an experiment could be performed, it would provide a population average that cannot address whether specific DNA tracts remain intact, are degraded, or get extended, so would not directly address roles of Rif1 in extension or restart. Combining the DNA combing with FISH would seem a better strategy for detailed analysis of effects and is one we are considering, but such an analysis goes beyond the scope of this study.

Also, why did the authors map track lengths (fibers) in Rif1 null cells when the entire paper is comparing WT to truncation mutant?

We have now included analysis of the effects of *rif1- $\Delta C594$* on nascent DNA protection, as shown in Fig 5C and EV4B and discussed on pages 10-11.

Most important, however, is the leap of logic from shorter replication fork tracks to the conclusion that Rif1 is preventing the degradation of recently synthesized DNA. Maybe Rif1 promotes replication restart, fork elongation rates - it seems that there are other interpretations that would require further investigation to sort out. I would recommend that the authors focus their conclusions on their intriguing findings that are clear and tone down the conclusions regarding resection and fork degradation.

Please see discussion above: it's not clear how differences in restart or elongation could explain the differences we see.

Other small comments:

- Figure 2B and 4E could use at least one more example.

We have added the gene MAM3 to Fig. 2B. We have moved the plot from original Fig. 4E to the Extended View Figures (Fig. EV4A), since all other examples were complicated by the existence of nearby peaks (e.g. caused by tRNA genes) and in any case it is unclear that we necessarily would

expect to see peak splitting, since forks will be distributed quite broadly around early origins in this experiment.

- Is there a difference between the fibers in Figure 5B?

Indeed there was no practical difference, so we have removed one picture and now show only one fibre image in Fig 5B.

- Truncated Rif1 binding is strikingly higher than wild type Rif1 at GAL4 upon induction, discussion of this result is lacking.

It may be that this result (shown in Fig. 6C of the original manuscript) reflects higher availability of the truncated mutant. But in any case we agree that this result is not fully consistent with expectations based on profiles seen in the ChIP-seq data. Therefore the *GAL1* model system is not necessarily faithfully representing effects of Rif1 and Rif1- Δ C594 association at other loci, and we have therefore removed this data from the manuscript (and shortened our discussion of the transcription-associated association).

- The authors conclude that there is 'a large portion of R-loops' overlapping with Rif1 binding sites but there is no evidence shown. Quantification of the overlap between Rif1 binding sites and R-loops/G4 motifs is lacking.

Although we have considerably shortened this section of the text we now provide a new table (Table EV2) showing the number of Rif1 ChIP peaks that coincide with positions of R loop-forming loci and predicted G4 sites.

Referee #2:

The Rif1 protein has emerged in several recent studies as a multifunctional, global regulator of genome replication and stability, in large part through its recently discovered role as a co-factor for the conserved protein phosphatase PP1. One of the roles for the Rif1-PP1 complex, conserved from yeast to humans, is to counteract potentially "premature" phosphorylation of the MCM helicase complex by S-phase activator DDK, and thus presumably premature origin activation. In addition, in yeast, Rif1 is also a Rap1 interacting protein and via this role contributes to telomere length control and transcriptional silencing.

This study begins with a high-resolution ChIP-Seq analyses of Rif1 binding in yeast, and succeeds in defining many new Rif1 target regions across the chromosome worth deeper mechanistic consideration in future studies. This represents the first high-resolution analyses of Rif1-chromosome target site binding on a genome-scale. Given Rif1's central importance in genome stability, these data are important to the field and of high potential physiological relevance. Thus this basic result, along with revealing a role for Rif1 at stalled replication forks (RF) in protecting nascent DNA, fits the stated mission of the EMBO Reports journal well: describing an interesting set of results of high potential physiological relevance without delving into the mechanisms underlying the outcomes.

There are some concerns about presentation/logic that I think could be dealt with by the authors with some editing. It's mostly about how some of the writing leads to confusion for the reader:

1. From the chromosome VI scan in Figure 1, excluding the telomeric regions, Rif1 and rif1DC594, a mutant version of Rif1 that prevents Rif1 from binding to Rap1, actually have very similar binding profiles across this chromosome don't they?. That is OK (and, I think, good news, actually)--these are the data, but this general observation does not mesh well with the logic of using the rif1DC594 mutant in the first place as it was presented. For example, line 28: "release of Rif1 from telomeres increases Rif1 binding to other genomic sites, revealing 100's of Rap1-dependent and Rap1-independent Rif1 sites"...This conclusion is not supported by the data in Figure 1, and so was quite confusing as a background rationale.

We have revised the text in the Introduction (e.g. page 4) and first part of the results (bottom of page 5) to emphasise the significance of analysis by ChIP-Seq (rather than by array) and to present the truncation mutant as used mainly to allow us to distinguish Rap1-dependent and independent peaks.

In terms of overall distribution of Rif1 target sites in Figure 1A, it simply appears that rif1DC594 produced a few weaker peaks compared to wild type Rif1, while it does not appear that many new sites were identified by using this mutant. While it is fair to conclude that you have identified many new Rif1 chromosomal target sites, I am not sure I am convinced that you reached that success because you used a rif1DC594 mutant. I think it is the ChIP_Seq technology.

We agree with R2's view that the improved depth of ChIP-Seq analysis was probably the most important factor in detecting peaks, and that while deleting the C-terminus was helpful it was actually not necessary. R2's comment is useful that the original manuscript text was overly focused on the historical reason for actually analysing the truncation mutant (as opposed to the interested results it in fact provided). We have re-configured our presentation of the mutant throughout the paper.

As I said, these are the data, but one reason I in some ways prefer this outcome is that I am not sure how interesting it would be to find target sites that only a rif1DC594 mutant could bind...

We agree, and now focus more on the general similarity of the binding profiles (except at sites where binding clearly is different, namely telomeres, centromeres, and around blocked forks).

2. Along these same lines, and related to the larger issue of determining the "no binding" chromosomal baseline for such experiments, the authors state: line 96: Rif1's strong association with Rap1 "tends to obscure Rif1 binding to other loci"....but that is simply not evident in the data--I see a lot of new "non-tel" sites for wild type Rif1 (blue) that do not seem to be obscured by Rif1's ability to bind Rap1 (because I see most of the same sites in rif1DC594 as well). The mutant and the wild type Rif1 have similar profiles. So from the data in Figure 2 it does not look as if releasing Rif1 from telomeres by inactivating its ability to bind Rap1 had all that much of an effect, other than reducing Rif1's association relative to the mutant at Rap1 sites (telomeres, ribosomal protein genes). How exactly were the binding data normalized in WT compared to the mutant?

The plotted signal reports enrichment at that site over what would be expected if all the reads were distributed randomly across the genome (corrected for the representation of sequence reads from each site in the Input sample). Thus the reviewer is correct that the 'baseline position' could be affected by the high number of reads for wt protein associated with telomeres, potentially making the peaks elsewhere appear smaller compared to the mutant.

How is the baseline (no binding) determined? Presumably the chromosomal baseline for WT, because so much binding occurs at telomeres, would be higher than that for the mutant and that could make it appear that the rif1DC594 binds some regions more effectively even if it does not--has a "correction" for that been made for the baselines in Figure 2 in WT vs mutant? If you generate baselines for both WT and mutant Rif1 where the telomeric regions (super strong Rif1-binding) are removed, does it now appear that WT Rif1 actually binds a lot better to more target sites than the mutant?

As suggested by the Reviewer, we did try re-normalising the results while excluding reads within 15 kb of a telomere. Unexpectedly, we found that this had the opposite effect from that which Reviewer 2 (and we) predicted: that is, peaks in the wild-type Rif1 protein actually became somewhat lower, when compared to enrichment found for the truncated protein. Otherwise, the plots were not greatly changed and the binding categories we discovered were unchanged. We have not changed the plots presented since the reasons for and interpretation of this effect are unclear, although they may relate to numbers of reads in the various samples. However, we have removed any conclusions based on different peak heights from the paper, and instead focus our interpretation simply on the rate at which peaks were identified at different types of site. We now mention the limitations on interpreting different peak heights in ChIP-Seq profiles (page 8, lines 208-210). Since the peak calling is done in a separate analysis using MACS2, it is not subject to the concerns outlined here.

How many sites showed greater association in WT Rif1 versus rif1DC594 and vice-versa?

Based on the issues above we have not attempted this analysis. Usefully comparing ChIP peak heights in genome-wide analysis is difficult for exactly the reasons that the Reviewer outlines, and is rarely done in this type of study because it is indeed difficult to be confident that the peak heights represent a biologically meaningful parameter.

It seems like strong conclusions were being made without data to support them in this instance.

We have removed conclusions based on 'increased' binding of the Rif1- Δ C594 mutant protein from the paper.

3. Rif1 binding to replication origins was clear and an important new piece of data. The differences between the behavior of Rif1 versus the rif1DC594 mutant at origins in cells arrested in HU was very interesting. The ability for Rif1 to interact with Rap1 is not needed for Rif1 to get to origins, but that C-terminal region that is nevertheless needed for the extended distribution of Rif1 around origins that are arrested. Interestingly though there was no effect on replication origin function as measured by BrdU IP between WT and rif1DC594 (Fig 4C)--Are blue and orange bars simply biological/independent replicates? I assume from the Figure legend yes, but why are they colored differently if there is no key in the figure indicating why?

Apologies for omitting this information from the Fig. 4 Legend in the original paper: they were indeed simply biological replicates. We now reformatted the data and presenting averages of biological replicates, as well as data points from each experiment (Fig 4C).

And if these represent replicates, why are they so different for rif1D at ARS607? It seems that the major result is how differently WT and the DC594 mutants act at origins arrested in HU and that point gets a little lost in some of the other details that seem less [did Reviewer's text get lost here?]

Actually the major point from this Figure is that the replication program appears very similar to WT in the *rif1- Δ C594* mutant, in that in HU an early origin (ARS607) has been activated while two different late origins remain inactive (in all strains but especially in *rif1- Δ C594* which looks essentially like wt). We have reconfigured the Figure to show early and late origins on the same scale to make this point more clear, and clarified the explanation in the text.

We agree that the actual IP efficiency percentage at ARS607 is somewhat variable, but the degree of variability is fairly typical for biological replicates in such experiments, and the difference is minor when compared to the difference from late origins.

4. The authors use DNA combing to test whether the broad association of Rif1 at origins arrested in HU that they observed might be relevant to stabilizing nascent DNA at stalled RFs, a phenotype that is emerging as relevant to Rif1 function in mammalian cells and so potentially extremely exciting. They found that indeed in rif1D cells nascent DNA was degraded suggesting that Rif1 in yeast functions to stabilize nascent DNA at stalled RFs. The issue that surprised me is that they didn't test what the rif1DC594 did in this assay and because they have the mutant and have been using it as a kind of control throughout the study this omission seemed odd, even though EMBO Reports does not require mechanism. Given that rif1DC594 does not show the same spread-out binding distribution as Rif1 at HU arrested forks but still binds to origins, this seemed like an obvious way to test whether that binding pattern of WT Rif1 was indeed relevant to nascent DNA protection.

R1 made the same point and we have now included analysis of the effects of *rif1- Δ C594* on nascent DNA protection, as shown in Fig 5C and EV4B and discussed on pages 10-11.

5. I was not convinced that the binding patterns of Rif1 over highly transcribed genes were not related to "hypechippability" artifacts based on the data shown even if the patterns differ between Rif1 versus rif1DC594--The no-tag control data that they report that they generated (as data not shown) should also be shown in Fig6. What kind of overlap did you observe between the ORFs you defined in this way and those discussed in the two relevant ChIP_Seq-Bias studies? Understanding how these types of signals can be generated, and their potential variation from experiment to experiment is

worth noting, regardless. In addition, there is an additional relevant reference besides that of Teytelman (Park, Lee, Bhupindersingh, Iyer, 2013, PLoS One)

We now cite this additional paper mentioned by Reviewer 2, and include 'untagged' control data in Fig. 6A. Rif1 and Rif1- Δ C594 are still enriched at highly transcribed genes when compared with our untagged, 'mock ChIP' data, as is clear from the plots and now mentioned in the text (pages 11 in Results and 15 in Discussion). There is indeed overlap between the Rif1 binding sites we identify in this category and 'hyperchippable' sites, which is inevitable as we are observing an association with highly transcribed genes. However based mainly on comparison with the untagged dataset, we do consider this category of Rif1 association to be genuine. Evidently though the meaning of this observation will remain unclear until a biological function for the association is shown, and accordingly we have shortened and softened the tone of this part of the paper, including removing original Fig. 6C (the GAL1 experiment) since its usefulness as a model was unclear.

Referee #3:

In their manuscript 'Budding yeast Rif1 binds to replication origins and blocked replication forks to protect nascent DNA' Hiraga et al. analysed the genome-wide binding sites of the budding yeast Rif1 protein, a factor involved in replication timing and telomere length control. Using a C-terminal deletion of Rif1 enabled them to investigate its role in a Rap1 independent manner.

In particular, Hiraga et al. show a most complete set of DNA associations in *S. cerevisiae* for RIF1, rif1 and rif1- Δ C594 and identified that Rif1 can bind a lot of genomic regions independent of Rap1 when deleted for aa594 onward. Interestingly, Rif1 and rif1- Δ C594 have different preferences for origin binding, which doesn't correlate with replication status/timing and no replication timing defect was observed for rif1- Δ C594. When treated in HU, deletion of Rif1 results in shorter nascent DNA tracks, suggesting a protective mechanism by Rif1. Finally, the authors found that Rif1- Δ C594 is recruited to highly transcribed class II and III genes.

Although some studies already identified the major binding sites of Rif1, this manuscript represents the most detailed description of the genomic binding sites of Rif1 (Rap1- dependent and - independent) under different growth conditions up to date. This manuscript provides a description of many new genomic Rif1 binding events and discusses their biological implication. Future work, as part of independent studies, will provide more evidence for each of the 6 classes of binding sites, but are not necessary for publication of this manuscript. Overall, the paper makes a very nice contribution towards the field and raises new hypotheses of Rif1 function in *S. cerevisiae* and beyond.

Major comments:

I. We suggest to add the data of Hafner et al., 2018, Cell Reports, which corroborates the findings of the authors. This gives the authors an opportunity to discuss similarities in the context of their 6 class model and differences they observed due to different techniques and mutants.

We now cover the Hafner paper in our Discussion (bottom of page 16). While the results of the papers are basically similar, Hafner et al do not describe some of our binding categories (such as blocked fork association and centromere association), partly because they did not include any cell cycle analysis.

One apparent inconsistency between the studies is that the Hafner et al paper argues that Rif1 binds preferentially to origins whose timing it regulates. However, since their data on this issue does not exclude telomeric and telomere-proximal origins, we suspect that this association primarily reflects the well-established effect of Rif1 on telomere replication timing, and in fact there is no discrepancy between the studies.

II. ChIP-qPCR: Please comment on how many biological replicates used in all relevant figures, why was PAC2 used as control gene? Its centromeric location (<15kb) could make it a suboptimal candidate gene, maybe the authors should use alternative control genes? The given values should be rather

described as average of biological replicates with standard deviation instead of median. Better information on error bar use throughout the manuscript. Fold enrichments below 1 may indicate unspecific enrichment of DNA (high background?)

In the original manuscript we used *PAC2* as a negative control as it had previously proved serviceable for studying Rif1 chromatin association to telomeres by ChIP-qPCR, where Rif1 exhibits strong association. However, at non-telomeric sites, it is perhaps true that *PAC2* is not the most appropriate negative control, as the reviewer suggests. Informed by the ChIP-Seq data, we instead designed a new primer pair at the gene *IRS4*, which gives lower background association, and have reanalyzed our ChIP-qPCR results in Fig. 4A with *IRS4* as a control as described in the Materials and Methods. The plots now show the average of two biological replicates, as well as each data point (each data point obtained based on three technical replicates).

Minor comments:

I. A graphical depiction of the 6 class model would be helpful for understanding.

We have added a graphical presentation illustrating our classes of Rif1 chromatin association as a Synopsis image.

II. Line 127: Figure legend and text appear not to fit in respect of permissive/semi-permissive/restrictive labelling or the text is unclear.

We have modified the text and now discuss 30 degrees as a 'restrictive' temperature.

III. FigS2/S5/S6: Colour code is misleading: Red usually implies high occupancies/ binding strength

We have reversed the colour code in this Figure as suggested.

IV. Fig.3C add labelling below as in B.

We have added the labelling as required.

V. Fig.4A: show more than one locus, early vs. late origin would be helpful

We now show ChIP-qPCR results at ARS1426 (an early origin) and at ARS1412 (late origin) in Fig 4A.

VI. Fig.4B: use a stacked column layout to make the differences in early vs. late origins more obvious

We have implemented this change to show the columns stacked.

VII. Fig.4B lacks a title

We have added the words 'Origin occupancy' as an annotation.

VIII. Fig. 4C lacks labelling of the orange and blue columns - it is unclear what is shown.

We now show the average of biological replicates, with a clear explanation in the legend.

IX. Fig.4D lacks a title and proper labelling

We have added the words 'Rif1 peak width' as an annotation.

X. Fig.5B: poor labelling, no title, maybe better micrographs available? Why is Rif1- Δ C594 not included?

We have clarified the explanation of this experiment in the text, added labelling to help readers, and included additional explanation in the figure legend. We now include nascent DNA stability data from rif1- Δ C594 mutant, in Fig 5C and in a replicate experiment in EV4B (discussed on pages 14-15 of the main text).

XI. Fig.S3C: the presented recruitment is not convincing, as recruitments are measured within the background

Instead of this now we include the result at an early origin (ARS1426) in Fig 4A.

XII. Fig.S4: same as in comment VI

We have made this change as suggested (now Fig EV2B).

XIII. Fig.6C: change y-axis labelling according to e.g. Fig.4A, add error bar for 'untagged +GAL' sample. How many biological replicates are shown?

We have removed these data, because the behaviour of Rif1 and Rif1- Δ C594 proteins differs in a way not generally reflective of the CHIP-Seq profiles, raising concerns that the GAL1 locus does not properly represent the association of these proteins with other genomic loci (see also Reviewer 2 second last point).

XIV. CHIP-qPCR primers and corresponding sequences are not included

This information is now presented in Table 2.

XV. Fig.S6: Rif1- Δ C594 lacks promoter associated peak, is this due to the inability to bind to Rap1? How does Rap1 behave at sites of high transcription?

Yes, Rif1- Δ C594 does not associate with this promoter presumably due to loss of Rap1 interaction, as explained in the Legend.

Rap1 has been reported to associate with promoters of some of highly transcribed genes (e.g., ribosomal proteins), but we are not aware of any study suggesting that Rap1 associates with coding sequences of highly transcribed genes in the way we describe for Rif1. Rif1 binding is unlikely to depend on Rap1 at such sites, since Rif1- Δ C594 binds highly transcribed loci in a similar way to full-length Rif1.

XVI. No p-values (significances as indicated by different amounts of *) explained in the manuscript as well as no indication on error bar usage throughout the manuscript.

Explanation now included in the Legend, and in other Legends throughout the manuscript.

We thank the Reviewers for their comments. They have improved the paper, which we hope is now ready for publication in EMBO Reports.

Thank you for the submission of your revised manuscript. We have now received the enclosed comments from referee 1 who has only minor concerns left, and I am happy to tell you that we can therefore in principle accept your manuscript.

In addition to addressing referee 1's concerns, a few other minor changes are needed:

- Please add all author contributions to the manuscript file, up to 5 keywords and a conflict of interest statement.
- Fig 3C is called out before Fig 3B, please correct.
- The figure panels with scale bars do not mention "n" as the number of independently performed experiments the data are based on, please add.

I have made a few minor changes to the abstract that needs to be written in present tense:

Despite its evolutionarily conserved function in controlling DNA replication, the chromosomal binding sites of the budding yeast Rif1 protein are not well understood. Here, we analyze genome-wide binding of budding yeast Rif1 by chromatin immunoprecipitation, during G1 phase and in S phase with replication progressing normally or blocked by hydroxyurea. Rif1 associates strongly with telomeres through interaction with Rap1. By comparing genomic binding of wild-type Rif1 and truncated Rif1 lacking the Rap1-interaction domain, we identify hundreds of Rap1-dependent and -independent chromosome interaction sites. Rif1 binds to centromeres, highly transcribed genes and replication origins in a Rap1-independent manner, associating with both early and late-initiating origins. Interestingly, Rif1 also binds around activated origins when replication progression is blocked by hydroxyurea, suggesting association with blocked forks. Using nascent DNA labeling and DNA combing techniques, we find that in cells treated with hydroxyurea, yeast Rif1 stabilizes recently synthesized DNA. Our results indicate that, in addition to controlling DNA replication initiation, budding yeast Rif1 plays an ongoing role after initiation and controls events at blocked replication forks.

Please let me know whether you agree with these changes.

When you upload a new manuscript version in our online system, you can bring forward all old files and then only replace the ones that need to be replaced.

REFEREE COMMENTS

Referee #1:

The authors have addressed all of my criticisms satisfactorily. Given the number of papers on the role of Rif1 in replication restart, I would urge them to be sure that they are satisfied that they have convinced readers this is unrelated. Regardless, the data are important to publish. A minor point is that they claim to have a Table EV2 in which they show overlap of Rif1 binding sites and R-loops, but that seems to be missing from the manuscript.

2nd Revision - authors' response

enter date

Thanks for your e-mail of 11 July accepting our manuscript in principle. We address the outstanding issues below.

> - Fig 3C is 5 called out before Fig 3B, please correct.

This seems to be an error: in the revised version Fig. 3B was first referred to on page 6, line 165 (in the merged PDF file), and Fig. 3C on page 7, line 173.

> - The figure panels with scale bars do not mention "n" as the number of independently performed experiments the data are based on, please add.

Presumably this comment refers to the box-whisker plots in Figs. 4D and 6B? We have clarified the notation of the plots in the Legends, and included mention of the numbers of origins and genes considered.

-Abstract

We have inserted your amendments to the Abstract in the new version

-Reviewer 1's further comment: Given the number of papers on the role of Rif1 in replication restart, I would urge them to be sure that they are satisfied that they have convinced readers this is unrelated.

> A minor point is that they claim to have a Table EV2 in which they show overlap of Rif1 binding sites and R-loops, but that seems to be missing from the manuscript.

Table EV2 was already included (perhaps Reviewer 1 didn't realise EV Tables are not in the main manuscript PDF).

We thank EMBO Reports for accepting our paper and for the speedy consideration.

3rd Editorial Decision

23rd Jul 18

I am very pleased to accept your manuscript for publication in the next available issue of EMBO reports. Thank you for your contribution to our journal.

At the end of this email I include important information about how to proceed. Please ensure that you take the time to read the information and complete and return the necessary forms to allow us to publish your manuscript as quickly as possible.

Thank you again for your contribution to EMBO reports and congratulations on a successful publication. Please consider us again in the future for your most exciting work.

Corresponding Author Name: Anne Donaldson

Manuscript Number: EMBOR-2018-46222V1